# A bromodomain–DNA interaction facilitates acetylation-dependent bivalent nucleosome recognition by the BET protein BRDT

Thomas C.R. Miller[1], Bernd Simon[1], Vladimir Rybin[1], Helga Grötsch[1], Sandrine Curtet[2], Saadi Khochbin[2], Teresa Carlomagno[1,3,4] & Christoph W. Müller[1]

Bromodomains are critical components of many chromatin modifying/remodelling proteins and are emerging therapeutic targets, yet how they interact with nucleosomes, rather than acetylated peptides, remains unclear. Using BRDT as a model, we characterized how the BET family of bromodomains interacts with site-specifically acetylated nucleosomes. Here we report that BRDT interacts with nucleosomes through its first (BD1), but not second (BD2) bromodomain, and that acetylated histone recognition by BD1 is complemented by a bromodomain–DNA interaction. Simultaneous DNA and histone recognition enhances BRDT's nucleosome binding affinity and specificity, and its ability to localize to acetylated chromatin in cells. Conservation of DNA binding in bromodomains of BRD2, BRD3 and BRD4, indicates that bivalent nucleosome recognition is a key feature of these bromodomains and possibly others. Our results elucidate the molecular mechanism of BRDT association with nucleosomes and identify structural features of the BET bromodomains that may be targeted for therapeutic inhibition.

[1] European Molecular Biology Laboratory (EMBL), Structural and Computational Biology Unit, Meyerhofstrasse 1, Heidelberg 69117, Germany. [2] CNRS UMR 5309, INSERM, U1209, Université Grenoble Alpes, Institut Albert Bonniot, 38700 Grenoble, France. [3] Leibniz University Hannover, Centre for Biomolecular Drug Research, D-30167 Hannover, Germany. [4] Helmholtz Centre for Infection Research, Group of Structural Chemistry, D-38124 Braunschweig, Germany. Correspondence and requests for materials should be addressed to T.C. (email: teresa.carlomagno@oci.uni-hannover.de) or to C.W.M. (email: christoph.mueller@embl.de).

The basic repeating unit of chromatin is the nucleosome, consisting of two copies each of four core histones (H2A, H2B, H3 and H4) wrapped in ~147 bp of DNA[1]. This repeating unit, and the higher order structures it forms, serves to regulate DNA accessibility for tight control of all DNA-templated processes. Access is mediated by epigenetic modifications including histone posttranslational modifications and DNA methylation, which either directly influence chromatin structure, or recruit or repel chromatin effector proteins that harbour modification-specific DNA or histone-binding modules, such as bromodomains[2–4].

The bromodomain and extra-terminal (BET) family (BRD2, 3, 4 and BRDT in human) are multi-functional chromatin effector proteins, whose critical roles in transcription and chromatin biology have made them attractive therapeutic targets for a wide range of malignancies (recently reviewed in refs 5–7). These proteins have a domain architecture that is conserved from yeast to human, which features two N-terminal bromodomains separated by a linker of ~110 amino acids, a 'motif B' that mediates BET protein dimerization[8], and a characteristic extra-terminal domain that acts as a protein–protein interaction module for recruiting cofactors involved in transcriptional regulation[9,10]. Additionally, longer isoforms of BRD4 and BRDT (Fig. 1a) have an extended C-terminus that allows them to facilitate RNA polymerase II-dependent transcription through interactions with the positive-transcriptional elongator complex[11,12].

Bromodomains recognize acetylated lysine residues and have a highly conserved structural fold consisting of four α-helices ($\alpha_Z$, $\alpha_A$, $\alpha_B$, $\alpha_C$) forming a compact left-handed bundle. Variable loops connecting helices $\alpha_Z$–$\alpha_A$ (ZA loop) and $\alpha_B$–$\alpha_C$ (BC loop) shape the acetyl–lysine binding pocket, thus contributing to substrate specificity[13,14]. Structure-based alignments and phylogenetic analysis of the 61 human bromodomains, which are found in 46 diverse proteins, divides them into eight distinct families[14]. The variability of the acetylated lysine pockets has recently allowed the development of inhibitors that are specific against various members of these families, particularly the BETs (recently reviewed in ref. 15).

Bromodomains typically bind to acetylated lysine residues with relatively low affinity (micromolar) and relatively poor selectivity for single acetylated-lysine residues within an isolated peptide[14]. However, specificity and affinity are frequently increased in the presence of multiple modifications. For example, both bromodomains of BRDT, the testis-specific member of the BET family that is essential for spermatogenesis, show a preference for multiply acetylated histone peptides[16,17]. The first bromodomain of BRDT (BD1) preferentially binds histone H4 tail peptides acetylated at lysines 5 and 8 ($H4K5_{ac}K8_{ac}$), while the second bromodomain (BD2) has highest affinity for histone H3 tail peptides acetylated at lysines 18 and 23 ($H3K18_{ac}K23_{ac}$). In both cases, binding to individually acetylated peptides is either weaker or could not be determined[16]. Notably, the crystal structure of the $BD1$-$H4K5_{ac}K8_{ac}$ peptide complex revealed that the single

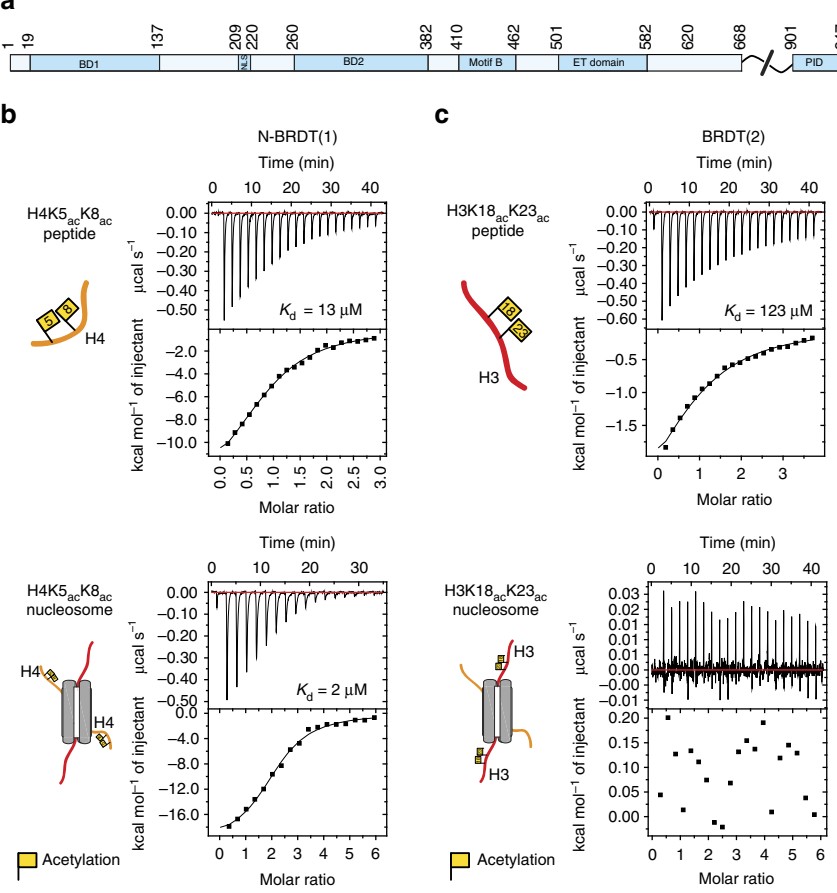

**Figure 1 | The nucleosome structure augments acetylated histone tail binding by BRDT-BD1 but prevents interaction with BRDT-BD2.** (**a**) Domain layout of human BRDT. (**b**) ITC profiles for N-BRDT(1) interactions with either acetylated histone H4 tail peptides or acetylated nucleosomes (both $H4K5_{ac}K8_{ac}$), as indicated. (**c**) ITC profiles for BRDT(2) interactions with either acetylated histone H3 tail peptides or acetylated nucleosomes (both $H3K18_{ac}K23_{ac}$), as indicated.

binding pocket of BD1 can simultaneously bind to both acetylated lysines of a $K5_{ac}K8_{ac}$ peptide[16].

There is increasing recognition that multivalency is a central component of chromatin biology. Chromatin-binding proteins and complexes frequently use multiple low-affinity interactions to achieve affinities, specificities and dynamics that would otherwise be impossible through monovalent interactions[18]. These multivalent interactions can occur either through the combination of histone or DNA interactions within a single protein, or within a complex carrying several chromatin-binding domains (recently reviewed in refs 19,20). Typically these interactions have been studied using a 'divide and conquer' approach, particularly in studies of histone tail binding by chromatin readers that have predominantly used isolated histone peptides in place of nucleosomes.

To gain a more comprehensive picture of how BET bromodomains interact with chromatin, we sought to characterize how BRDT, with its tandem bromodomains, interacts with acetylated nucleosomes. Using a combination of biophysical methods, including isothermal titration calorimetry (ITC) and methyl-TROSY NMR, we find that BRDT interacts with acetylated nucleosomes through BD1 only, while BD2 is unexpectedly unable to interact with acetylated histones within a nucleosome. We show that BRDT-BD1 binding to nucleosomes is bivalent and consists of simultaneous recognition of both acetylated histone tails and DNA. Importantly, we find that the bromodomain–DNA interaction is nonspecific and facilitates recruitment of BRDT to bulk chromatin, where it may assist BRDT in bringing about chromatin compaction in response to histone hyperacetylation in cells. We find similar DNA binding in other members of the BET bromodomain family, suggesting that this bivalent mode of recruitment is not limited to BRDT. Our results offer an in-depth characterization of bromodomain association with acetylated nucleosomes and highlight the importance of nucleosome components beyond the histone tail for bromodomain recognition.

## Results

**Nucleosome structure influences bromodomain binding**. The interactions of the BET bromodomains with chromatin have largely been studied by two approaches. First, chromatin immunoprecipitation experiments and pull-downs have been used to correlate BET binding with the presence of particular histone modifications on chromatin *in vivo*[14,21]. Alternatively, the bromodomains have been tested individually by *in vitro* experiments, such as peptide arrays and ITC, to investigate their binding preferences for specifically modified histone tail peptides[14,16]. Here we extend these studies and use BRDT to characterize the interaction of the BET bromodomains with *in vitro* reconstituted, site-specifically acetylated nucleosomes.

On the basis of previous characterization of the binding specificities of the two bromodomains of murine Brdt[16], we utilized a histone semisynthesis approach and native chemical ligation to reconstitute four types of acetylated nucleosomes *in vitro*[22,23]. These nucleosomes were either unmodified, acetylated on histone H4 ($H4K5_{ac}K8_{ac}$), acetylated on histone H3 ($H3K18_{ac}K23_{ac}$) or acetylated on both H3 and H4. The histone modifications did not interfere with octamer or nucleosome refolding (Supplementary Fig. 1) and therefore were used to investigate how nucleosome structure affects bromodomain binding to histone tails.

We tested two BRDT bromodomain constructs for their histone- and nucleosome-binding properties. The first construct (N-BRDT(1), residues 1–143) included the core bromodomain fold of BD1 and the N-terminus of BRDT that has been shown to

be essential for BRDT's ability to compact acetylated chromatin[24]. The second construct, (BRDT(2), residues 258–383) was based on the murine Brdt-BD2 bromodomain crystal structure[16].

As expected, ITC showed that N-BRDT(1) requires acetylation of histone H4 for binding to histone peptides and nucleosomes. No binding was detected for unmodified samples (Supplementary Fig. 2a,b), whereas the dissociation constants between N-BRDT(1) and $H4K5_{ac}K8_{ac}$ acetylated peptides and nucleosomes were found to be $13\,\mu M$ and $2\,\mu M$, respectively, (Fig. 1b). These data show that BRDT-BD1 interacts with nucleosomes with a $>6$-fold enhancement in affinity compared with acetylated H4 histone tail peptides alone, and indicates that the nucleosome structure is an important determinant of histone tail recognition.

Surprisingly, our ITC data indicated that BRDT(2) cannot interact with acetylated nucleosomes, despite interacting with the equivalently acetylated histone H3 peptides (Fig. 1c). This suggests that the nucleosome structure prevents BRDT(2) from binding the H3 tail.

To confirm our ITC data and further characterize how BRDT interacts with nucleosomes, we adopted the methyl-TROSY NMR methodology that facilitates the investigation of large complexes[25]. Here $^{13}C$, $^1H$ methyl groups of isoleucine, leucine and valine in a perdeuterated background were used as sensitive probes for monitoring interactions between the bromodomains and acetylated nucleosomes.

Consistent with our ITC data, N-BRDT(1) interacted with ($H4K5_{ac}K8_{ac}$) acetylated nucleosomes (Fig. 2a), whereas BRDT(2) showed no interaction with $H3K18_{ac}K23_{ac}$ acetylated nucleosomes (Supplementary Fig. 2c). For N-BRDT(1), overlaid $^{13}C$-$^1H$ methyl-TROSY spectra showed chemical shift perturbations (CSPs) in residues surrounding the acetylated lysine-binding pocket when tested with both ($H4K5_{ac}K8_{ac}$) acetylated peptides and nucleosomes (Fig. 2a,b). In addition, binding of N-BRDT(1) to acetylated nucleosomes also induced specific CSPs distinct from those caused by peptide binding alone (Fig. 2a,b). These perturbations occurred towards the opposite end of N-BRDT(1) (relative to the histone-binding pocket), and suggest that BD1 may make additional contacts with the nucleosome outside of the histone H4 tail.

**BRDT-BD2 is tethered to acetylated nucleosomes by BD1**. We hypothesized that BRDT-BD2 may require BRDT-BD1 for recruitment to acetylated nucleosomes and thus may only bind when tethered by BD1. Therefore, we expressed and purified a longer BRDT construct encompassing both bromodomains (BRDT(1-2)) to verify whether both BD1 and BD2 interact with acetylated nucleosomes when linked.

We first characterized this construct by size-exclusion chromatography, analytical ultracentrifugation, NMR spectroscopy and small angle X-ray scattering (SAXS) (Supplementary Fig. 3) and found that BRDT(1–2) is a monomeric, elongated and flexible molecule with the two bromodomains at either end of an unfolded linker (Supplementary Fig. 3). Comparison of overlaid $^1H$, $^{15}N$ HSQC spectra (Supplementary Fig. 3c) and $^{13}C$-$^1H$ methyl-TROSY spectra (Supplementary Fig. 4a) of BRDT(1–2) with individual N-BRDT(1) and BRDT(2) data showed a good correspondence between peak positions. Our experimental data therefore give no indication of significant changes in the folds of these domains when linked, nor of dimerization between them, as has been proposed for other bromodomains of the BET family[8,26,27]. New resonances, which appeared in the BRDT (1–2) $^1H$, $^{15}N$ HSQC spectrum (Supplementary Fig. 3c), were attributed to the linker; these resonances were predominantly grouped in the middle of the spectrum and have significantly

 

higher intensities, suggesting that the linker is unstructured. Comparable peak intensities for the resonances of the bromo-domains in the linked construct, when compared with those of N-BRDT(1) and BRDT(2) alone, indicate that the two domains rotate independently of each other.

We then tested how BRDT(1–2) interacts with acetylated peptides and nucleosomes using leucine and valine $^{13}$C-$^{1}$H methyl-TROSY NMR. Overlaid spectra of samples containing BRDT(1–2) and acetylated H3 or H4 histone peptides show CSPs for both BD1 and BD2 (Supplementary Fig. 4a, lower panels). The observed CSPs occur in almost identical resonances regardless of the peptide, as would be expected for both peptides targeting the histone-binding pockets. These data therefore show that both bromodomains bind to both acetylated histone peptides and highlight the lack of specificity of individual bromodomains for acetylated histone peptides alone.

In contrast to the peptide binding, we found that only BD1 was able to interact with nucleosomes uniformly acetylated on both histones H3 (K18$_{ac}$K23$_{ac}$) and H4 (K5$_{ac}$K8$_{ac}$) (Supplementary Fig. 4b, lower panels). Resonances from the linker and BD2 did not show evidence of interaction with the nucleosomes and appear to have remained flexible in solution. This result shows that tethering of the BD2 to nucleosomes is not sufficient to induce interaction.

To confirm that the experimental conditions were conducive to seeing binding between BD2 and nucleosomes, we repeated the experiments with acetylated histones from the purified octamer used for nucleosome reconstitution. BRDT(1–2) binding was tested under the same concentrations and buffer conditions as the nucleosome experiments (Supplementary Fig. 4b, upper panels).

Here BD2 interacted with the histones, demonstrating that BD2 is unable to bind to acetylated H3 tails when presented in the context of a nucleosome.

**BRDT interacts non-specifically with DNA through BD1.** Given the proximity of the histone H4 tail to the DNA that encircles the histone octamer, we investigated whether BRDT-BD1 might interact with DNA using electrophoretic mobility shift assays (EMSAs). Strikingly, N-BRDT(1) showed robust binding to the 167 bp Widom DNA that we use for reconstituting nucleosomes (15 µM) (Fig. 3a), whereas BRDT(2) did not interact with the DNA (Supplementary Fig. 5).

To investigate the binding specificity and stoichiometry of these interactions, we tested N-BRDT(1) binding to 66 and 25 bp DNA oligonucleotides (Fig. 3b). These oligonucleotides had an unrelated sequence to the previously tested 167 bp DNA (sequences in Methods). N-BRDT(1) interacted with each of the DNAs, showing that it interacts with nucleic acids without sequence specificity. Quantification of the EMSAs showed that N-BRDT(1) interacts with 66 and 25 bp DNAs with dissociation constants of 29 µM and 52 µM, respectively. An apparent increase in affinity as the DNA length increases is expected for a non-specific DNA-binding protein presented with an increased number of potential binding sites in longer DNA. Accordingly, the shifts seen in the EMSAs suggest that at least two N-BRDT(1) molecules can interact with 66 bp DNA, whereas only a single shift is seen for 25 bp DNA (Fig. 3b, arrowheads).

Our ITC data (Fig. 1b; Supplementary Fig. 2a,b) suggested that N-BRDT(1) required acetylation of H4 to bind to nucleosomes,

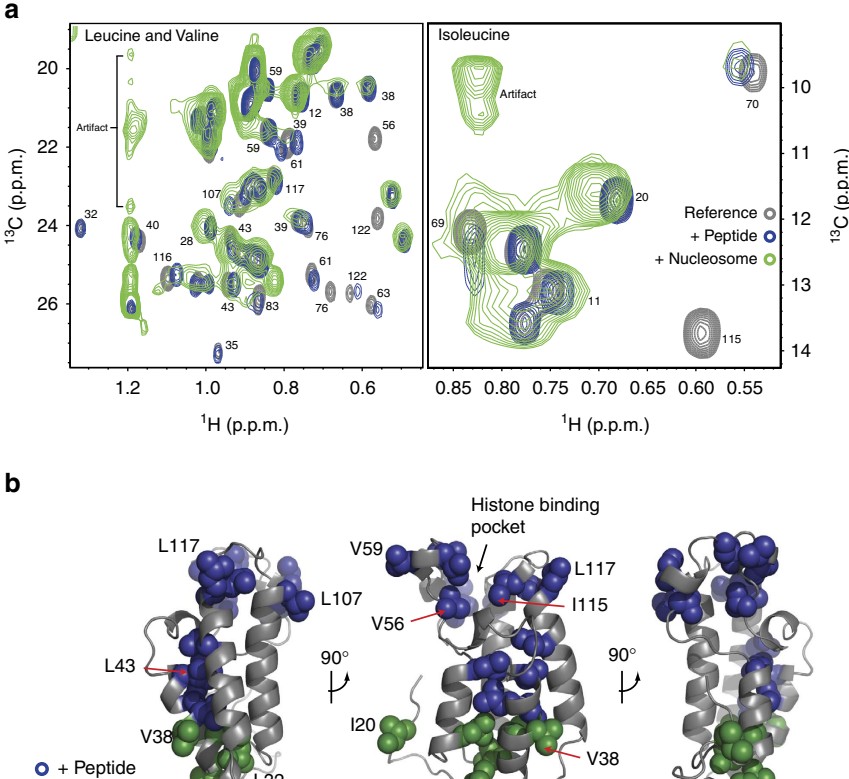

**Figure 2 | BRDT interacts bivalently with nucleosomes through BD1.** (**a**) Overlaid $^{13}$C-$^{1}$H methyl-TROSY spectra of isoleucine, leucine and valine labelled N-BRDT(1) alone or in the presence of H4K5$_{ac}$K8$_{ac}$ peptides or H4K5$_{ac}$K8$_{ac}$ nucleosomes. (**b**) Residues showing CSPs in **a** upon binding of nucleosomes (blue and green) or peptides only (blue) displayed on a homology model of human BRDT-BD1 (generated using the Phyre2 web server[28]).

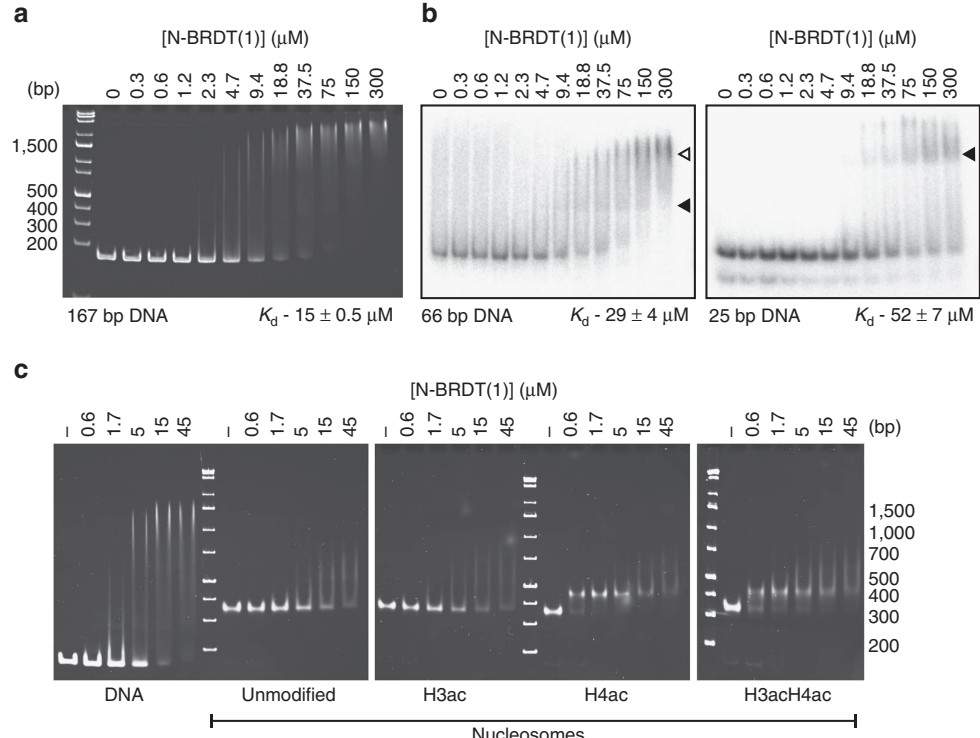

**Figure 3 | BRDT interacts non-specifically with DNA through BD1.** (**a**) EMSA of a titration of N-BRDT(1) incubated with 167 bp double-stranded DNA containing a centralized Widom 601 sequence[49]. DNA (0.2 μM) was mixed with N-BRDT(1) (0.3–300 μM) in a final volume of 6 μl and incubated for 30 min before native-PAGE electrophoresis (4 °C) and visualization with ethidium bromide staining. Unbound DNA was quantified using a Typhoon imager with ImageQuant software and the obtained data used to calculate the binding affinity indicated. (**b**) EMSA titrations of N-BRDT(1) interacting with either 66 or 25 bp radio-labelled DNA, as indicated. Shifted bands containing BRDT(1)-DNA complexes with one (filled arrowhead) or two (unfilled arrowhead) N-BRDT(1) molecules are indicated. Affinities shown were calculated from quantification of unbound DNA as detailed in **a**. (**c**) EMSA titrations of N-BRDT(1) interacting with either 167 bp Widom DNA, unmodified nucleosomes or nucleosomes carrying acetylated H3(K18$_{ac}$K23$_{ac}$) and/or H4(K5$_{ac}$K8$_{ac}$) tails, as indicated.

however, binding of N-BRDT(1) to free DNA would suggest that it should also bind non-specifically to nucleosomes, independent of acetylation. To test this, we performed EMSA experiments with 167 bp Widom DNA and unmodified or acetylated nucleosomes (Fig. 3c). We found that N-BRDT(1) did indeed show weak binding to unmodified nucleosomes in a similar manner seen for DNA. The apparent discrepancy with the ITC result suggests that—due to limitations in the achievable sample concentrations—the interaction could not be detected under our experimental conditions.

As predicted from our ITC experiments (Fig. 1b), N-BRDT(1) binds with a significantly higher affinity and apparent specificity to nucleosomes containing acetylated histone H4 (Fig. 3c). In contrast, BRDT(2) showed no interaction with unmodified or acetylated nucleosomes (Supplementary Fig. 5), supporting our findings from NMR and ITC experiments.

These data demonstrate DNA binding by a bromodomain and show that BRDT-BD1 interacts non-specifically with DNA and nucleosomes. BRDT therefore has a significantly different mode of interacting with nucleosomes than previously envisaged, and may target bulk chromatin through low affinity, non-specific DNA interactions, priming it to interact tightly and specifically following histone hyperacetylation.

**BD1 interacts with DNA and H4 through distinct interfaces.** To further characterize the BD1–DNA interaction, we analyzed the sequences of human BRDT BD1 and BD2 (Supplementary Fig. 6a), the X-ray crystal structure of BD1 (2RFJ[14]) and a homology model of BD2 (generated using the Phyre2 web server[28]) (Fig. 4a). These analyses identified a positively charged patch in BD1 but not BD2 that correlates in location with residues showing NMR CSPs specific to nucleosome binding (Fig. 2a,b). We speculated that this region may be responsible for interacting with DNA and may explain the difference in DNA binding between BD1 and BD2.

The positively charged patch of BD1 is centered on the first α-helix (αZ) and features three prominent lysine residues (K37, K41 and K45) (Fig. 4a). We mutated these three lysines to serine both individually and in combination to test their effect on DNA binding.

We recorded 1D NMR spectra to ensure that the mutations had not affected the folding of the bromodomains and additionally tested each of the mutated proteins by ITC for binding to H4K5$_{ac}$K8$_{ac}$ peptides (Supplementary Fig. 6b,c). Neither mutation of the central lysine, nor triple mutation significantly altered the 1D NMR spectra or the acetylated histone H4 peptide-binding affinity in ITC. This is in contrast to a previously characterized point mutation in the histone-binding pocket (I155Y) that, despite maintaining the bromodomain fold, significantly reduced the binding affinity of N-BRDT(1) for acetylated histone peptides from 13 to >600 μM (Supplementary Fig. 6b,c).

Interestingly, individual mutation of each lysine reduced N-BRDT(1) binding to DNA, with the most pronounced effect seen by mutating the central lysine (K41S) (Fig. 4b). Double mutation of lysine residues 37 and 41 (2KS), or triple mutation of all three (3KS) abolished DNA binding (Fig. 4b).

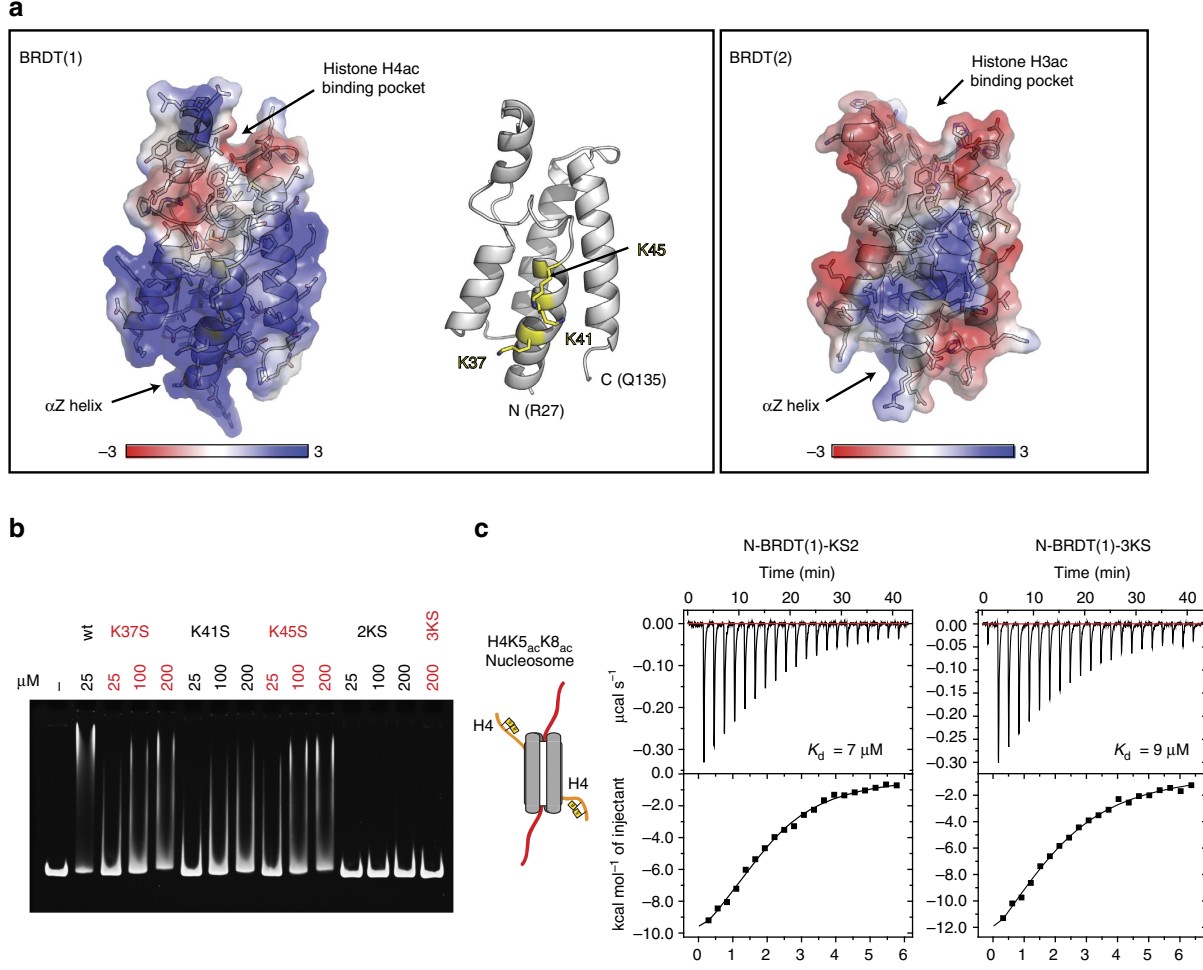

**Figure 4 | Residues involved in BRDT-BD1 DNA binding are distinct from those required for histone H4 recognition.** (**a**) Crystal structure of human BRDT-BD1 (PDB code – 2RFJ[14]) and homology model of human BRDT-BD2 (generated using the Phyre2 web server[28]), as indicated. Structures are shown as cartoons with side chain sticks and partially transparent electrostatic surfaces generated using PBD2PQR and APBS[56–58]. The positively charged patch centers on the αZ helix, which contains lysines K37, K41 and K45 (highlighted in yellow). (**b**) EMSA of N-BRDT(1) lysine mutants (concentrations as indicated) binding to 167 bp double-stranded DNA (1 μM). Mutant '2KS' is a double mutant K37S, K41S. Mutant '3KS' contains all three point mutations. (**c**) ITC profiles for N-BRDT(1) mutants interactions with H4K5$_{ac}$K8$_{ac}$ acetylated nucleosomes, as indicated.

The bromodomain is a small domain and given the spatial proximity between the histone-binding pocket and DNA-binding interface, we speculated that DNA binding may be influenced by histone tail binding. We performed quantitative EMSA titrations comparing N-BRDT(1) binding to 66 bp DNA +/− H4K5$_{ac}$K8$_{ac}$ peptides. Peptide binding did not significantly affect N-BRDT(1)'s affinity for 66 bp DNA (33 μM and 29 μM, respectively; Supplementary Fig. 7), showing that the two binding sites function independently. Therefore, the increased binding affinity between BRDT-BD1 and nucleosomes, when compared with DNA or histone peptides alone, appears to occur largely through the entropic benefits of bivalency[18], rather than through allostery or the direct effect of a composite binding interface, such as seen for the chromodomain of Chp1 (ref. 29).

Next, we tested the affinity of our N-BRDT(1) DNA binding mutants for H4K5$_{ac}$K8$_{ac}$ nucleosomes, to quantify the role of DNA binding in nucleosome recognition (Fig. 4c). The single point mutation of the central lysine (K41S) reduced nucleosome-binding affinity ∼3.5-fold (7 μM), while triple mutation of all three lysines further reduced the affinity to 9 μM; approaching the affinity of N-BRDT(1) for H4K5$_{ac}$K8$_{ac}$ peptides alone (∼13 μM) (Fig. 1b). These results show that the BD1–DNA interaction contributes to

nucleosome-binding affinity. BRDT-BD1 thus interacts bivalently with nucleosomes via a combination of acetylation-specific histone recognition and non-specific DNA binding.

**Nucleosome structure influences bromodomain specificity.** We have demonstrated that BRDT-BD1 interacts with both acetylated histone tails and DNA simultaneously to augment its nucleosome-binding affinity; however, we also wanted to address the issue of whether nucleosome structure contributes to binding specificity. Bromodomains are promiscuous for binding to acetylated lysine residues within a variety of sequence contexts[14]. Although multiple acetylations within a single peptide enhance both the specificity and affinity of BET bromodomain binding, some (particularly BRD4-BD2) are still able to bind to a large variety of multiply acetylated histone peptides[14].

To investigate whether nucleosome structure contributes to target specificity, we produced chimeric nucleosomes in which the acetylated histone H4 tail (H4K5$_{ac}$K8$_{ac}$) was ligated to the core of histone H3. This allowed us to test the binding affinity of N-BRDT(1) to nucleosomes containing the optimal acetylated histone H4 recognition motif (H4K5$_{ac}$K8$_{ac}$), but in an alternative position on the nucleosome.

Our ITC data indicate that N-BRDT(1) interacts with chimeric nucleosomes with a dissociation constant of $5\,\mu M$ (Supplementary Fig. 8). Notably, this represents an $\sim 2.5$-fold enhancement in binding affinity over the interaction between N-BRDT(1) and the $H4K5_{ac}K8_{ac}$ peptide, but an $\sim 2.5$-fold decrease in affinity compared with $H4K5_{ac}K8_{ac}$ nucleosomes (Fig. 1b). We propose that DNA binding localizes the bromodomain to the chimeric nucleosome in the same manner as for the wild type (WT) nucleosome, thus enhancing the local concentration and affinity. However, in the case of the WT nucleosome, the position of the acetylated histone tail relative to the acetyl–lysine-binding pocket (orientated by the DNA) is more favourable compared with the same acetylated sequence in the chimeric location. Therefore, the affinity for the WT nucleosome is higher and an additional layer of specificity is generated.

This result shows that the peptide sequence surrounding acetylated lysines is not the only determinant of bromodomain specificity. Rather, the nucleosome structure and, specifically, the position of the acetylated lysines relative to DNA also influence bromodomain recognition. This demonstrates that the specificity of a chromatin reader can only accurately be evaluated with nucleosomes and/or chromatin and cannot be reliably judged at the level of peptides.

**Other BET bromodomains interact with DNA**. To investigate whether DNA binding is a general feature of the BET bromodomains or is specific to BRDT-BD1, we inspected the electrostatic surface charges of the human BET bromodomains and found that each member (with the exception of BRDT-BD2) also contains a positively charged patch (Fig. 5a). Although the residues involved are not identical to those in BRDT-BD1, the positional conservation of this patch suggests that this region may also interact with DNA. We therefore tested the other bromodomains for DNA binding by EMSA (Fig. 5b). Interestingly, the first bromodomain of BRD2 and the second bromodomains of BRD2, 3 and 4 all interacted with DNA, demonstrating that bromodomain-mediated DNA binding is a conserved feature among all members of the BET family.

Surprisingly, however, the first bromodomains of BRD3 and BRD4 did not interact with DNA, despite the presence of the positively charged face. This result shows that inspection of the electrostatic surface potentials alone is not adequate for predicting an interaction between the bromodomains and DNA. The positive charge of this surface is therefore necessary but not sufficient for DNA interaction and specific residues and/or conformation must also be important.

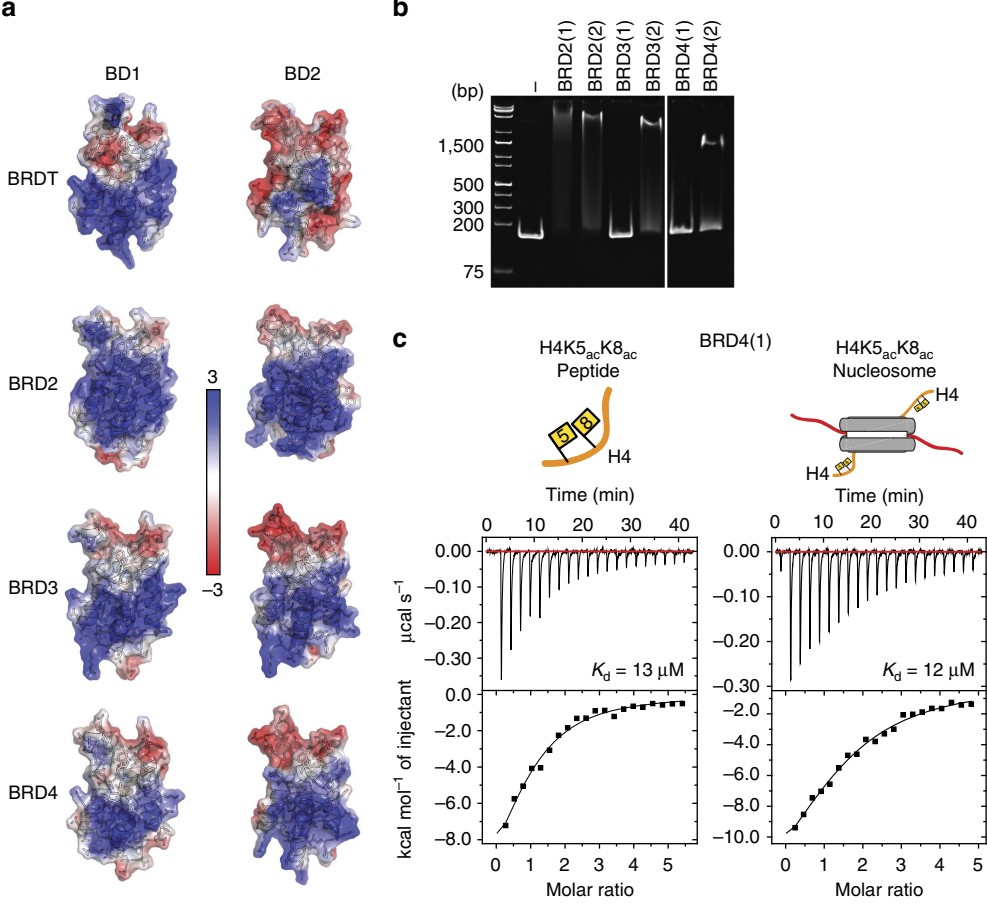

**Figure 5 | Other BET bromodomains interact with DNA.** (**a**) Structures of the human BET bromodomains (PDB codes: BRDT-BD1 (2RFJ); BRD3-BD1 (2NXB); BRD3-BD2 (2OO1); BRD4-BD1 (2OSS, N-terminally truncated to R58 for comparison with other BD1 structures); BRD4-BD2 (2OUO)[14]; BRD2-BD1 (1X0J)[26]; BRDT-BD2 structure is a homology model generated using the Phyre2 web server[28]; BRD2-BD2 (2DVV)). Structures are shown as cartoons with side chain sticks and partially transparent electrostatic surfaces generated using PBD2PQR and APBS[56–58]. (**b**) EMSA of BET bromodomains (as indicated) interacting with 167 bp double stranded DNA. DNA ($0.5\,\mu M$) was mixed with BET bromodomains ($100\,\mu M$) in a final volume of $4\,\mu l$ and incubated for 30 min before native-PAGE electrophoresis at $4\,^{\circ}C$ and visualization with ethidium bromide staining. (**c**) ITC profiles for BRD4(1) interactions with $H4K5_{ac}K8_{ac}$ peptides and nucleosomes, as indicated.

Given the difference we find in the DNA binding ability of BRD4-BD1 and BRDT-BD1, we wondered if this was reflected in their nucleosome-binding affinities. We therefore compared BRD4-BD1 binding to acetylated nucleosomes and histone peptides by ITC (Fig. 5c). Interestingly, in contrast to BRDT-BD1, BRD4-BD1 does not bind acetylated nucleosomes with higher affinity than acetylated histone peptides. This supports our hypothesis that the BRDT-BD1–nucleosome interaction is bivalent and is enhanced by simultaneous interaction with acetylated histone H4 and DNA. BRD4-BD1 does not interact with DNA, thus, the binding affinities for histone peptides and nucleosomes are the same.

**BD1-DNA binding helps recruit BRDT to chromatin in cells**. Histone binding by BRDT-BD1 is essential for BRDT's ability to compact chromatin in an acetylation-dependent manner[16,24]. To investigate whether BD1-mediated DNA binding has a similar functional significance, we assayed WT and mutant BRDT constructs for their ability to localize to, and compact, hyperacetylated chromatin when ectopically expressed in cells. On the basis of equivalent studies on murine Brdt[16,24], we cloned a GFP-tagged human BRDT construct (aas 1–444; ΔC-sBRDT) that is predicted to be highly active in chromatin compaction. We then made triple lysine to glutamate (3KE) mutations that abolished N-BRDT(1) DNA binding (Fig. 6a), without interfering with acetylated histone peptide recognition (Fig. 6b). ITC data indicate that N-BRDT(1)-3KE has a fivefold reduced affinity for H4K5$_{ac}$K8$_{ac}$ acetylated nucleosomes when compared with WT N-BRDT(1) (Fig. 6b).

Fluorescence recovery after photobleaching (FRAP) experiments show that WT human and murine ΔC-sBRDT constructs have similar fluorescence recovery half-lives ($T_{1/2} = 5$–7 s) following trichostatin A (TSA)-induced histone hyperacetylation (Fig. 6c; Supplementary Fig. 9). This suggests that the human protein interacts with acetylated chromatin in a similar manner to the murine protein. In contrast, human and murine ΔC-sBRDT proteins with mutations in their histone-binding pockets (I115Y and P50A/F51A/V55A (PFV1)[16,24], respectively), and a DNA-binding deficient human protein (ΔC-sBRDT-3KE) showed a significantly faster recovery half-life ($T_{1/2} = \sim 1.5$ s); indicative of reduced association with acetylated chromatin (Fig. 6c; Supplementary Fig. 9). Double mutation of both the histone and DNA-binding interfaces of human ΔC-sBRDT (3KE/I115Y) did not further decrease the fluorescence recovery half-life. Mutation of either site alone is therefore sufficient to prevent acetylated chromatin association in cells (Fig. 6c; Supplementary Fig. 9). Control FRAP experiments with and without TSA confirm that human ΔC-sBRDT 3KE, I115Y and double mutants are similarly unresponsive to TSA-induced histone hyperacetylation (Supplementary Fig. 10).

In accordance with the findings above, human ΔC-sBRDT-3KE, I115Y and the double mutant are also compromised in their ability to compact TSA-induced hyperacetylated chromatin in cells, when compared with the WT protein (Fig. 6d). Instead, ΔC-sBRDT with the 3KE and/or I115Y mutations remains diffuse in the nucleus following TSA treatment. These results support our finding that BRDT-BD2 cannot associate with nucleosomes; showing that only a fully functional BD1, capable of bivalent recognition of both DNA and acetylated histone tails, is able to recruit BRDT to acetylated chromatin.

## Discussion
The BETs are a highly conserved protein family involved in diverse functions including transcriptional regulation and chromatin remodelling. Through their bromodomains, they are known to interact with acetylated histone tails and have a rare ability to stay associated with chromatin throughout the cell cycle[30–32]. Here we have used BRDT as a model for studying the interactions of the BET bromodomains with acetylated nucleosomes and find that DNA plays a critical role.

BRDT-BD1 interacts with doubly acetylated histone H4 tails with a low micromolar affinity and associates with chromatin in vivo in an acetylation-dependent manner[16,24]. We find that BRDT-BD1 also interacts with DNA and this interaction stimulates the binding of the bromodomain to acetylated nucleosomes in vitro and acetylated chromatin in cells. Manual alignment of a Brdt-BD1-H4K5$_{ac}$K8$_{ac}$ peptide crystal structure alongside the structure of the nucleosome (1AOI[1]) suggests how binding affinity and specificity may be enhanced (Supplementary Fig. 11). Orientation of BD1 with its DNA-binding interface towards the DNA positions the acetyl–lysine-binding pocket in the correct orientation for interaction with the histone H4 tail. The distance of $\sim 20$ Å between H4R17 leaving the surface of the nucleosome and the pre-aligned BD1 acetyl–lysine pocket would be bridgeable for the extended histone tail, allowing BD1 to recognize H4K5$_{ac}$K8$_{ac}$.

Under this scenario, the non-specific interaction with DNA would serve to localize BD1 to nucleosomes and would allow BRDT to scan chromatin for acetylated histone H4. Stable BD1 binding would then be dependent on recognition of the appropriate acetylated lysine residues. BRDT would thus be stabilized on nucleosomes in a bivalent manner, similar to that of the PWWP domain of LEDGF, which simultaneously binds H3K36me3 and DNA to enhance its nucleosome-binding affinity[33,34].

A corollary of this model is that the relative position/spacing of acetylated lysines—relative to the nucleosome core—would influence bromodomain target specificity. We were able to demonstrate that this is the case by showing that BRDT-BD1 binds to WT and H3-chimeric nucleosomes containing acetyl–lysine residues in identical sequence contexts, but different positions, with differing affinities. Thus, the nucleosome structure imposes an added layer of specificity to bromodomain binding.

In stark contrast to BRDT-BD1, and predictions on the basis of peptide-binding studies, we find that BD2 neither interacts with DNA nor with acetylated nucleosomes. Whereas conservation of the contiguous positively charged patch on the rest of the BET bromodomains implies conservation of function, BRDT-BD2 has a considerably more negative electrostatic surface potential (Fig. 5a). This may actively repel BRDT-BD2 from DNA, thus preventing an interaction with acetylated histone H3 tails in the context of the nucleosome, even when BRDT-BD2 is physically tethered to the nucleosomes by BRDT-BD1 binding to acetylated H4.

Although nucleosome binding can be enhanced by tandem domains bivalently binding nucleosomes (2–3-fold enhancement for bromo-PHD of BPTF[35]; 3–11-fold enhancement for two PHD fingers of CHD4 (ref. 36)), this is not the case for BRDT. Unlike the structured spacer element that connects the PHD and bromodomain of BPTF[35], or the relatively short linker between the PHD fingers of CHD4 (ref. 36), we find that the linker connecting the bromodomains of BRDT is long ($\sim 110$ aa) and disordered (Supplementary Fig. 3). Furthermore, it remains so upon BRDT binding to nucleosomes (Supplementary Fig. 4b). With this long flexible linker, the two domains are effectively independent in solution, precluding any binding enhancement through 'prepaying' entropic costs of positioning the domains for binding[18]. Despite this, tethering of BRDT to nucleosomes by BD1 would still increase the relative local concentration of BD2 and would therefore be expected to increase the likelihood of an interaction, if it were not inhibited. This has been shown to be the case for BRD4, where a construct encompassing BRD4-BD1 and

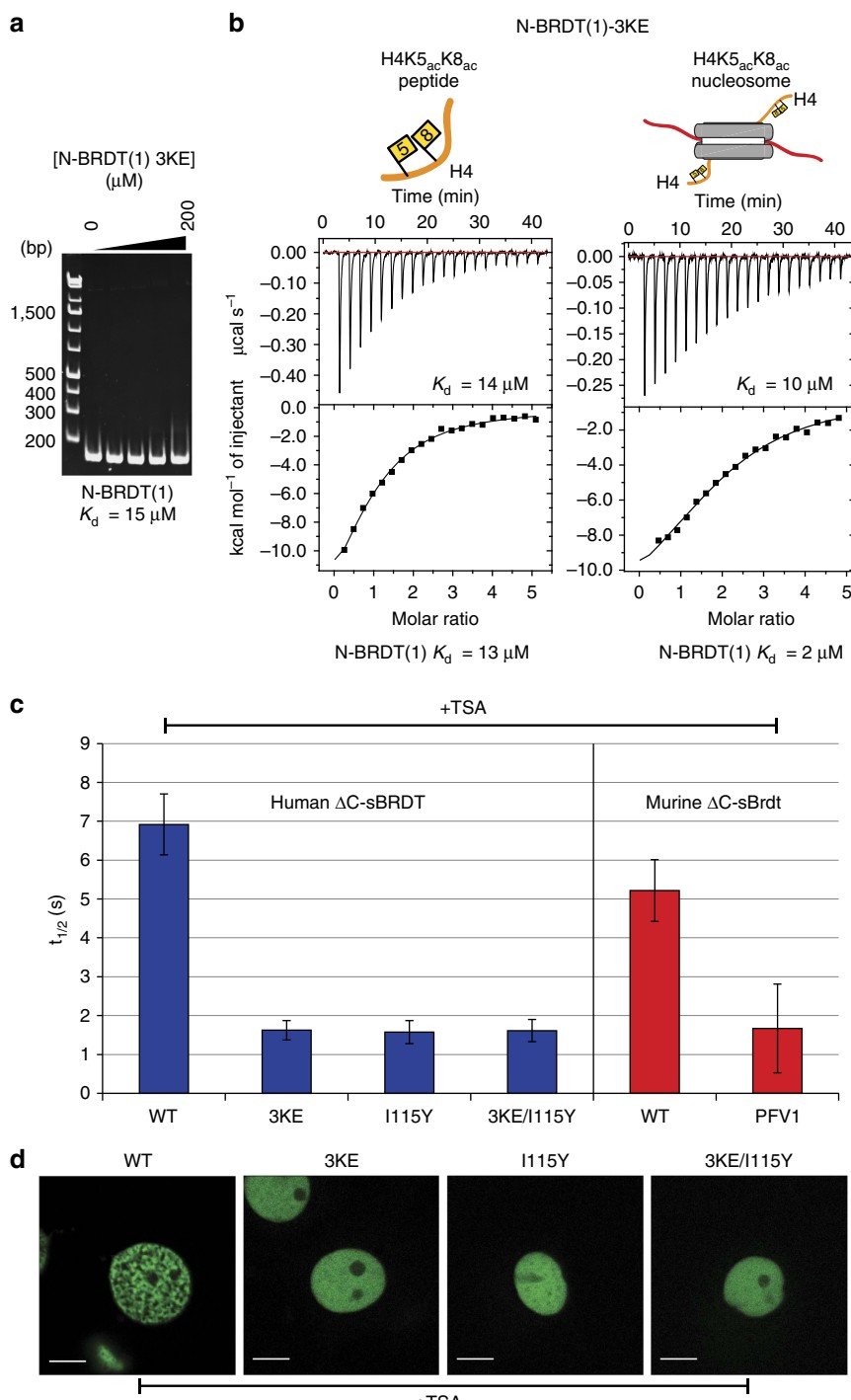

**Figure 6 | DNA binding by BD1 is important for BRDT localization to chromatin in cells.** (**a**) EMSA of N-BRDT(1) lysine mutants (concentrations: 3, 12.5, 50 and 200 μM) binding to 167 bp double-stranded DNA. DNA (0.4 μM) was mixed with N-BRDT(1) 3KE in a final volume of 5 μl and incubated for 30 min before native-PAGE electrophoresis at 4 °C and visualization with ethidium bromide staining. (**b**) ITC profiles for N-BRDT(1) 3KE interactions with either acetylated histone H4 tail peptides or acetylated nucleosomes (both H4K5$_{ac}$K8$_{ac}$), as indicated. (**c**) FRAP analysis of human and murine ΔC-sBRDT constructs in the presence of TSA-induced histone hyperacetylation. Cos7 cells were transfected by vectors expressing GFP-tagged WT and mutant ΔC-sBRDT constructs (as indicated) and cells were treated with the histone deacetylase inhibitor TSA (100 ng ml$^{-1}$) to induce histone hyperacetylation. A decrease in fluorescence recovery half-life ($t_{1/2}$) indicates an increase in protein mobility, and reduced chromatin association. The indicated fluorescence recovery half-lives are mean values obtained from 10 independent cells ($n = 9$ for WT murine ΔC-Brdt). Error bars show the s.e.m. (**d**) Confocal microscopy images of representative transfected cells following TSA treatment (Scale bars, 10 μm).

BD2 was found to bind to nucleosomes acetylated on both histones H3 and H4 with a 2.6-fold increased affinity over BRD4-BD1 alone[37].

Previous research has shown that BRDT localization and its ability to compact chromatin in somatic cells are primarily mediated by BRDT-BD1, while BRDT-BD2 is, at least in part,

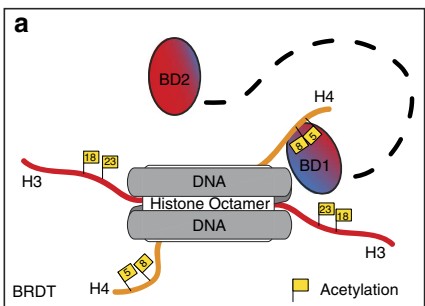 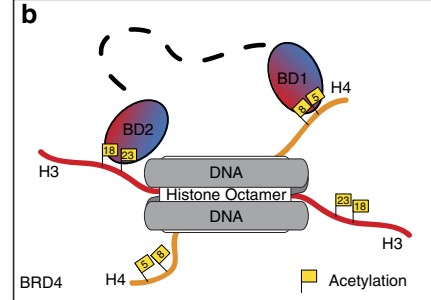

**Figure 7 | Schematic models of BRDT and BRD4 bromodomains interacting with acetylated nucleosomes. (a)** BRDT interacts with acetylated nucleosomes via its BD1 domain. Binding may be initiated through non-specific interactions with DNA, which allow BRDT to localize to chromatin. Specificity is generated through recognition of tandem acetylated lysine residues ($K5_{ac}/K8_{ac}$) on the histone H4 tail, while the affinity of the interaction is enhanced by bivalent interaction with both the histone tail and DNA. In contrast, BRDT-BD2 does not interact with acetylated nucleosomes and therefore is flexibly tethered to nucleosomes via BRDT-BD1. BRDT-BD2 may function to recruit as-yet-unknown acetylated non-histone proteins to the chromatin. **(b)** BRD4 interacts with acetylated nucleosomes via both its BD1 and BD2 domains. Our results indicate that BRD4-BD1 binds to nucleosomes through the acetylated histone H4 tail and does not additionally interact with DNA. Bivalent binding of BRD4 through both bromodomains has previously been shown to enhance BRD4 binding affinity for nucleosomes by 2.6-fold[37].

dispensable[16,24]. Our results give a molecular explanation to the observed biological data and indicate that BRDT-BD2 may have an alternate biological function. One possible function, which would depend on flexible tethering of BD2 to nucleosomes by BD1, would be the recruitment of an acetylated non-histone protein to chromatin. BRDT would not be unique in this function; other BET proteins are also known to use their bromodomains for interactions with acetylated non-histone proteins[38–42].

Possible candidates for BRDT-BD2 recruitment would be the transition proteins (TPs) and protamines (Prms); both of which are known to be acetylated[43–45], and both of which are critical for BRDT-mediated post-meiotic genome repackaging during spermatogenesis[12]. TPs and Prms depend on BRDT for nuclear localization in elongating spermatids, and loss of BD1 leads to their accumulation in the cytoplasm, preventing histone replacement[12]. Although BRDT has a nuclear localization sequence, this data indicates that chromatin binding by BRDT-BD1 is required for nuclear retention of BRDT, TPs and Prms and therefore that BRDT interacts, either directly, or indirectly, with TPs and Prms—potentially via BRDT-BD2.

In contrast to BRDT-BD1 and BD2, BRD4-BD1 has unambiguously been shown to bind to histone peptides[14,31], mononucleosomes[37] and chromatin[31], but also to other non-histone acetylated proteins (reviewed in ref. 5). We find that BRD4-BD1 does not interact with DNA, which led us to compare BRD4-BD1 and BRDT-BD1 binding to acetylated nucleosomes by ITC. Although BRDT-BD1 and BRD4-BD1 have similar binding affinities for $H4K5_{ac}K8_{ac}$ peptides, their binding affinities for nucleosomes differed significantly ($\sim$6-fold). This is largely because BRD4-BD1 does not show any enhancement of binding affinity for nucleosomes over isolated peptides.

On the basis of our data, we propose three models for how BET bromodomains interact with nucleosomes (Fig. 7). BRDT-BD1 has a non-specific DNA-binding activity, which allows positioning of the bromodomain on the nucleosome for interaction with acetylated histone H4 tails. This bivalent interaction thus enhances the binding affinity and specificity of BRDT-BD1 for nucleosomes (Fig. 7a). BRDT-BD2 does not interact with DNA or nucleosomes (Fig. 7a), likely due to electrostatic repulsion from the DNA preventing BRDT-BD2 from accessing the histone H3 tail. Thus, it cannot enhance the binding of BRDT to nucleosomes. Finally, BRD4-BD1 retains a positively charged interface on a polarized surface but does not interact with DNA with an affinity measurable in our experiments. This domain

interacts with $H4K5_{ac}K8_{ac}$, however, it shows no enhancement of binding stimulated by the nucleosome context. Unlike BRDT, BRD4 binding to nucleosomes is enhanced by simultaneous binding of both of its bromodomains[37] (Fig. 7b).

Like BRD4-BD1, we show that BRD3-BD1 does not interact with DNA, while BRD2-BD1, BRD2-BD2, BRD3-BD2 and BRD4-BD2 do. In future work it will be interesting to assess the importance of these findings in nucleosome binding, transcriptional regulation and chromatin remodelling by the BET proteins *in vivo*. Furthermore, the BET bromodomains are just 8 of 61 human bromodomains. Our work provides a demonstration of DNA binding by a member of the bromodomain family; it will be intriguing to see whether bivalent binding of DNA and acetylated histone tails by bromodomains is a conserved feature of the wider bromodomain family of chromatin 'reader' modules.

In summary, we have used BRDT as a model to study the binding of BET bromodomains to site-specifically acetylated nucleosomes. ITC experiments revealed that BRDT-BD1 has an enhanced affinity for acetylated nucleosomes over acetylated histone peptides, whereas BRDT-BD2 does not interact with acetylated nucleosomes, in contrast to predictions on the basis of peptide-binding studies. Using a range of biophysical methods and mutational analyses, both *in vitro* and in cells, we show that BRDT-BD1 bivalently interacts with acetylated nucleosomes and chromatin through concomitant interaction with histone H4 and DNA. We show that bromodomain-mediated DNA binding is conserved among members of the BET family, indicating that bivalent nucleosome recognition through simultaneous DNA and histone tail binding is an important component of nucleosome recognition by BET bromodomains and possibly in bromodomains beyond the BET family. Our results emphasize the importance of studying chromatin reader's interactions with nucleosomes rather than isolated peptides or DNA. Accordingly, this study provides important insight into the molecular mechanism of BET association with chromatin and shows that features outside of the bromodomains' histone-binding pockets are crucial for interactions with nucleosomes. Knowledge of these features may pave the way for enhanced targeting of specific BET bromodomains for therapeutic purposes.

## Methods
**Expression and purification of human BET constructs.** BRDT constructs N-BRDT(1) (aas 1–143), BRDT(2) (aas 258–383) and BRDT(1–2) (aas 1–383) were cloned using restriction-free cloning[46] as N-terminally His-TEV-tagged fusion

proteins into pETM11 expression vectors. Original sequences were amplified from cDNA clones provided by Sino Biological Inc (catalogue number: HG11602-M).

BET bromodomains BRD2(2) (aas 348–455), BRD3(1) (aas 24–144), BRD3(2) (aas 306–416), BRD4(1) (aas 44–168) and BRD4(2) (aas 333–460) were gifts from Nicola Burgess-Brown provided through Addgene (plasmid numbers: 53626, 38940, 38941, 38942 and 38943, respectively). Codon-optimized BRD2(1) (aas 48–184; Eurofins) was cloned using restriction-free cloning[46] as an N-terminally His-TEV-tagged fusion protein into the pETM11 expression vector.

Plasmids were transformed into the *Escherichia coli* strain BL21-CodonPlus (DE3)-RIL (Stratagene), and the bacteria were grown in Luria-Bertani (LB) medium supplemented with $50 \, \mu g \, ml^{-1}$ kanamycin and $25 \, \mu g \, ml^{-1}$ chloramphenicol at 37 °C. Expression of recombinant proteins was induced by addition of 0.4 mM IPTG at an $OD_{600}$ of 0.6 followed by overnight incubation at 18 °C. The bacterial cell pellets were resuspended in lysis buffer (50 mM Tris (pH 8.0), 500 mM NaCl, Complete EDTA-free Protease Inhibitor Cocktail (Roche), DNase1 (Roche), 0.25 mM DTT) before being lysed by sonication and clarified by centrifugation at 75,600g. Proteins were purified from the soluble fraction using nickel-NTA agarose (Qiagen), followed by TEV protease cleavage of the N-terminal His-tag and additional purification over Ni-NTA resin to remove the tags and His-tagged TEV protease. Finally, proteins were further purified by size exclusion chromatography using Superdex 75 or 200 columns (GE Healthcare) as appropriate. The purity and structural integrity of the purified proteins was monitored by SDS–PAGE and $^1H$ NMR (Supplementary Fig. 12).

**NMR labelled BRDT constructs.** For NMR experiments, labelled proteins were obtained by expression in M9 media (containing 100% $D_2O$ for uniformly deuterated proteins); $^{13}C$ and $^{15}N$ labelling was achieved through the addition of $^{13}C$ glucose ($4 \, g \, l^{-1}$) and $^{15}NH_4Cl$ ($1 \, g \, l^{-1}$), while specific $^{13}C$, $^1H$ labelling of ILV-methyl groups was achieved through the addition of labelled sodium salts of labelled alpha-ketoisovaleric acid (1,2,3,4-$^{13}$C4; 3,4',4',4'-D4 for N-BRDT(1); 3-methyl-$^{13}$C; 3,4,4,4-D4 for BRDT(2) and BRDT(1–2); $120 \, mg \, l^{-1}$) and alpha-ketobutyric acid ($^{13}$C4; 3,3-D2 for N-BRDT(1); methyl-$^{13}$C; 3,3-D2 for BRDT(2) and BRDT(1–2); $60 \, mg \, l^{-1}$) when the cells reached an $OD_{600}$ of 0.6. Protein expression was induced by IPTG (0.4 mM) and carried out overnight at 18 °C. This protocol was on the basis of the protocol developed by the Kay laboratory[47]. NMR isotopes were obtained from Cambridge Isotope Laboratories. Proteins were purified as described for the unlabelled equivalents, with a final dialysis step into NMR buffer (25 mM sodium phosphate (pH 6.7), 100 mM NaCl) prepared with $H_2O$ or $D_2O$ as appropriate.

**Nucleosome preparation.** *Full-length unmodified histones.* Histones H2A/H2B–Codon-optimized (supplied by Entelechon – now Eurofins), full-length human histones H2A/H2B were co-expressed from a pCDF-DUET vector transformed into the *E. coli* strain BL21-CodonPlus (DE3)-RIL (Stratagene). Bacteria were grown in LB medium supplemented with $100 \, \mu g \, ml^{-1}$ streptomycin and $25 \, \mu g \, ml^{-1}$ chloramphenicol at 37 °C and induced at an $OD_{600}$ of 0.6 with IPTG (0.25 mM). Cells were collected after 4 h by centrifugation. Cell pellets were resuspended in cold lysis buffer (20 mM Tris (pH 8.0), 100 mM NaCl, 1 mM EDTA, 10 mM β-mercaptoethanol and Complete EDTA-free Protease Inhibitor Cocktail (Roche)) and the cells lysed using an Emulsiflex-C3 homogenizer (Avestin) and clarified by centrifugation at 75,600g. Clarified cell lysate was filtered using a 0.45 μm syringe filter (Merck Millipore) and injected onto combined 5 ml HiTrap Q HP and HiTrap Heparin HP columns (GE Healthcare) pre-equilibrated in lysis buffer. Columns were washed with one column volume of lysis buffer before detaching the HiTrap Q HP column. The HiTrap Heparin HP column was subsequently washed with 20% elution buffer (20 mM Tris (pH 8.0), 2 M NaCl, 0.1 mM EDTA, 10 mM β-mercaptoethanol) before eluting the H2A/H2B complex with a 20 CV gradient into 100% elution buffer. The pooled and concentrated fractions containing the H2A/H2B dimer were further purified by size exclusion chromatography over a Superdex 75 column. Protein purity was monitored by SDS–PAGE. The H2A/H2B complex was extensively dialysed into 1 mM DTT before lyophilisation and storage at − 80 °C.

Histones H3 and H4 – Codon-optimized (supplied by Entelechon – now Eurofins), full-length human histone H3 and histone H4 (kindly provided by T.Bartke) were individually expressed from pETM-13 and pETM-21b( + ) vectors transformed into the *E. coli* strain Rosetta (DE3) pLysS (Novagen). Bacteria were grown in LB medium supplemented with $50 \, \mu g \, ml^{-1}$ kanamycin (H3) or ampicillin (H4) and $25 \, \mu g \, ml^{-1}$ chloramphenicol at 37 °C and induced at an $OD_{600}$ of 0.6 with IPTG (0.25 mM). Cells were collected after 4 h by centrifugation. Histones were purified essentially as previously described[48]. Cell pellets were resuspended in cold histone wash buffer (50 mM Tris (pH 7.5), 100 mM NaCl, 1 mM EDTA and Complete EDTA-free Protease Inhibitor Cocktail (Roche)). Cells were lysed using an Emulsiflex-C3 homogenizer (Avestin) and inclusion bodies pelleted by centrifugation at 12,000g. Pellets were resuspended in cold histone wash buffer + 1% Triton-X-100 in a Dounce homogenizer and inclusion bodies again pelleted by centrifugation at 12,000g. Inclusion body washing was repeated twice with Triton-X-100, before two final washes into histone wash buffer. Inclusion bodies were resuspended in unfolding buffer (20 mM Tris (pH 7.5), 7 M guanidine hydrochloride and 100 mM DTT) for 1 h at room temperature, before clarification by centrifugation at 20,000g. Supernatants were filtered using a 0.45 μm syringe

filter (Merck Millipore) and histones purified by size exclusion chromatography over a Superdex 200 column (GE Healthcare) in SAU-1000 buffer (7 M urea, 20 mM sodium acetate (pH 5.2), 1 M NaCl, 1 mM EDTA and 5 mM β-mercaptoethanol). Fractions containing histone H3 or H4 were pooled and diluted to bring the NaCl concentration to below 200 mM and loaded on to a HiTrap SP HP column (GE Healthcare) equilibrated in SAU-200 (as above, with 200 mM NaCl). Histone H3 was eluted using a gradient into SAU-600 buffer (as above, with 600 mM NaCl). Histone H4 was eluted using a gradient into TU-1000 buffer (7 M urea, 20 mM Tris (pH 7.5), 1 M NaCl, 1 mM EDTA and 5 mM β-mercaptoethanol). Protein purity was monitored by SDS–PAGE. The H3 and H4 histones were extensively dialysed into 1 mM DTT before lyophilisation and storage at − 80 °C.

**Truncated histones H3 and H4 for native chemical ligation.** Codon-optimized human histones H3 and H4 (supplied by Entelechon – now Eurofins) were used for cloning H3(A25C) and H4(A15C) mutants into a pETM13 vector using restriction-free cloning[46]. These constructs contain an N-terminal methionine initiation codon immediately followed by the alanine-to-cysteine mutation, thus lack residues 2–24 and 2–14, respectively. The *E. coli* methionyl-aminopeptidase removes the N-terminal methionine thus exposing the cysteine at the N-terminus. Plasmids were transformed into the *E. coli* strain Rosetta (DE3) pLysS (Novagen), and bacteria were grown in LB medium supplemented with $50 \, \mu g \, ml^{-1}$ kanamycin and $25 \, \mu g \, ml^{-1}$ chloramphenicol at 37 °C. Cells were induced at an $OD_{600}$ of 0.6 with IPTG (0.25 mM) and collected after 4 h by centrifugation. The truncated histones were purified as described for full-length H3 but in the absence of reducing agents. Purified histones were dialysed into water and lyophilized.

**Native chemical ligation.** For ligations, truncated histones H3(A25C) and H4(A15C) were incubated with acetylated histone peptides carrying a C-terminal thioester in ligation buffer (200 mM NaPO4 (pH 7.5), 0.5 mM TCEP, 6 M guanidine HCl, 100 mM sodium 2-mercaptoethanesulfonate) for 24 h at 25 °C. The reactions were stopped by adding DTT to a final concentration of 100 mM.

Ligated H3 was diluted in SAU-0 (7 M urea, 20 mM sodium acetate (pH 5.2), 1 mM EDTA, 5 mM β-mercaptoethanol) to dilute the guanidine HCl concentration to < 200 mM and the protein was loaded on to a HiTrap SP HP column (GE Healthcare) pre-equilibrated in SAU-200 (200 mM NaCl). A 5 CV gradient to 25% SAU-600 (600 mM NaCl) followed by 10 CV at 25% eluted the vast majority of unligated H3. Ligated H3 was eluted with a 5 CV gradient to 50% SAU-600 followed by a step to 100% SAU-600 to remove any remaining protein. Protein purity was checked by 18% SDS–PAGE and impure fractions were pooled for a second round of purification.

Truncated H4(A15C) and H3(A25C) (for chimeric nucleosomes) were ligated to a peptide consisting of a His-tag, TEV-cleavage site and the H4 N-terminal sequence (1–14) and therefore could be purified using Ni-NTA agarose (Qiagen). Ligated H4 was extensively dialysed into binding buffer (100 mM NaPO4, 10 mM Tris, 6 M guanidine HCl, pH adjusted to 8.0) and then incubated with pre-equilibrated Ni-NTA resin. Ligated H4 was eluted with elution buffer (100 mM NaPO4, 10 mM Tris, 8 M urea, pH adjusted to 4.5) and dialysed into 1 mM DTT before lyophilisation. Protein purity was checked by 18% SDS–PAGE.

**167 bp Widom DNA.** A plasmid containing 80 repeats of 167 bp with a centered Widom 601 sequence[49] was amplified in XL1 Blue cells (Stratagene). Plasmid DNA was purified by Gigaprep (Qiagen) using the standard protocol. Purified plasmid was digested using AvaI (NEB) to isolate 167 bp repeats, which were purified from the vector backbone by size exclusion chromatography with an XK 16/70 Superose 6 pg (GE Healthcare) gel filtration column. Purified DNA was precipitated by addition of 0.7 volumes of isopropanol and 0.3 M sodium acetate and pelleted by centrifugation at 20,000g for 1 h at 4 °C. DNA was resuspended in TE buffer (10 mM Tris (pH 8.0), 1 mM EDTA) and the purity checked on a 2% agarose gel.

**Nucleosome reconstitution.** Histone octamers were refolded from purified histones and assembled into nucleosomes with 167 bp DNA by salt deposition[48]. Histone aliquots were dissolved in unfolding buffer (20 mM Tris (pH 7.5), 6 M guanidine hydrochloride and 20 mM DTT), before being mixed in equimolar ratios and diluted to give a final protein concentration of $1 \, mg \, ml^{-1}$. The histones were then dialysed at 4 °C against refolding buffer (10 mM Tris (pH 7.5), 2 M NaCl, 1 mM EDTA and 5 mM β-mercaptoethanol). Refolded octamers were concentrated and then purified by size exclusion chromatography over a Superdex 200 column (GE Healthcare). Purified histone octamers were mixed with 167 bp Widom DNA in refolding buffer and diluted to a final DNA concentration of $0.7 \, mg \, ml^{-1}$. The samples were dialysed into buffers containing 20 mM Tris (pH 7.5) 5 mM β-mercaptoethanol and 1 mM EDTA, with decreasing concentrations of NaCl (2 M, 850 mM, 650 mM and 150 mM). Reconstitution conditions were optimized by titration and nucleosomes checked by 5% native PAGE. N-terminal His-tags on ligated H4 of H4K5K8ac nucleosomes and H3-H4K5acK8ac chimeric nucleosomes were removed by TEV cleavage (2 h, 30 °C) of reconstituted nucleosomes (Supplementary Fig. 1). Cleaved nucleosomes were purified by incubation with Ni-NTA agarose and collection of flow-through. Chimeric nucleosomes contained

chimeric H3-H4K5$_{ac}$K8$_{ac}$ and WT H2A, H2B and H4, and thus contained both modified and unmodified H4 tail sequences.

**Isothermal titration calorimetry.** ITC was carried out at 20 °C with an ITC 200 Microcalorimeter (GE Healthcare) following dialysis of purified BRDT proteins and nucleosomes into interaction buffer (20 mM Tris (pH 8.0), 150 mM NaCl, 1 mM EDTA and 1 mM TCEP). For BRDT and BRD4 BD1 peptide-binding experiments, histone peptides (450 μM) were titrated into BD1 proteins (25–30 μM). For BRDT-BD2 peptide-binding experiments, histone peptides (2.5 mM) were titrated into BRDT-BD2 proteins (90 μM). For nucleosome-binding experiments, BET bromodomain proteins (200–260 μM) were titrated into nucleosomes (6–11 μM). Peptides were supplied by Peptide Protein Research Ltd and were solubilized in interaction buffer. ITC data was analyzed with using the MicroCal Origin software package after correction for heats of dilution. A summary of all ITC presented in this paper can be found in Supplementary Table 1.

**Nuclear magnetic resonance.** NMR experiments were performed on Bruker Avance III 600 and 800 MHz spectrometers equipped with HCN triple-resonance cryo-probes. Protein assignments were obtained using a combination of standard triple resonance experiments[50]. Nucleosome interaction was monitored by $^{13}$C-$^{1}$H HMQC experiments recorded for 20 h on labelled BRDT constructs titrated into acetylated histones or nucleosomes (10 μM) at 34 °C. NMR data were processed by NMRPipe[51] and analyzed with NMRView[52].

**Electrophoretic mobility shift assays.** For the binding reaction, a master mix containing DNA and reaction buffer (20 mM Tris (pH 7.5), 100 mM NaCl, 5 mM DTT, 0.5 mM EDTA) was prepared and mixed with dilutions of BRDT constructs to the final concentrations indicated. Binding was performed for 30 min at 4 °C. BRDT-DNA complexes were resolved by PAGE (5–7% polyacrylamide, 3% glycerol and 0.5 × TAE) at 100 V for ~1.5 h (4 °C). Experiments utilizing 167 bp DNA were stained with ethidium bromide and imaged using the AlphaImager HP imaging system (Proteinsimple). EMSA gels using $^{32}$P-labelled 25 and 66 bp dsDNA oligos were dried and exposed to a phosphorimager screen. Unbound DNA was quantified as a proportion of total signal/lane using a Typhoon imager with ImageQuant software and data plotted against protein concentration to calculate the binding affinities indicated.

**DNA sequences used for EMSA-binding studies.** *167bp Widom DNA.* Forward—5′-tcgggggccgccctggagaatcccggtgccgaggccgctcaattggtcgtagacagctctagcac cgcttaaacgcacgtacgc-3′ 5′-gctgtccccgcgtttaaccgccaaggggattactccctagtctccaggcacgt gtcagatatatacatcctgtgcatgtac-3′

Reverse—5′-ccgagtacatgcacaggatgtatatatctgacacgtgcctggagactagggagtaatcccttgg cggttaaaacgcggggggacagc-3′ 5′-gcgtacgtgcgtttaagcggtgctagagctgtctacgaccaattgagcggcc tcggcaccgggattctccagggcgccc-3′

*66 bp DNA.* 5′-cgatataagtgtaacggctatcacatcacgctttcaccgtggagaccgggggttcgact ccccgtatc-3′

*25 bp DNA.* 5′-cgaaagtggccgagtggtctatggcg-3′.

**Fluorescence recovery after photobleaching.** FRAP experiments were performed as previously described[16]. In brief, 1.5 μg of each of the GFP-BRDT constructs was transfected into Cos7 cells using lipofectamine 2000. Cells were treated with TSA (50 ng ml$^{-1}$) and incubated at 37 °C in 5% CO2 for 16 h. FRAP analysis was performed using a Zeiss microscope (LSM710 NLO-LIVE7-Confocor3) equipped with a 488 nm laser and a LP505 filter, on 10 independent cells. A circular region was bleached for 1.22 s; fluorescence recovery curves were individually fitted with the ZEN software using the single exponential model $I(t) = A(1 - e^{-\frac{t}{T_1}})$, where $I(t)$ represents the fluorescence intensity as a function of time $t$ and $A$ is the mean post-bleached fluorescence signal. Recovery times ($t_{1/2}$) were determined using $t_{1/2} = T1\ln(2)$ for each dataset individually and used to calculate the mean. Cos7 cells (ATCC CRL1651) were authenticated and supplied by the American Type Culture Collection (ATCC). This cell line has been tested for mycoplasma contamination with the MycoAlert Mycoplasma Detection Kit (Lonza).

**Analytical ultracentrifugation.** Analytical centrifugation was perform in a Beckman Optima XL-A centrifuge fitted with AN-60 rotor and double-sector aluminum centerpieces (40,000 r.p.m.; 129,000$g$; 4 °C). Sedimentation velocity profiles were recorded at 280 nm. Molecular weight distributions were determined by the C(s) method implemented in the Sedfit software[53].

**Small angle X-ray scattering.** SAXS data of BRDT(1-2) were acquired at BM29 at the ESRF[54], Grenoble with protein concentrations of 0.5, 1, 2 and 4 mg ml$^{-1}$ in a buffer containing 25 mM tris (pH 8) and 200 mM NaCl supplemented with 2 mM dithiothreitol (DTT) to reduce radiation damage. Measurements were carried out at 20 °C with samples exposed for 10 frames of 1 s each at full transmission. The

data were analyzed with the ATSAS package[55]. The final curve was generated by extrapolating the data to zero concentration.

**Data availability.** All relevant data reported in this paper are available from the authors upon reasonable request. The following PDB structures were used in this study: human BET bromodomains BRDT-BD1 (2RFJ), BRD3-BD1 (2NXB), BRD3-BD2 (2OO1), BRD4-BD1 (2OSS), BRD4-BD2 (2OUO), BRD2-BD1 (1X0J) and BRD2-BD2 (2DVV).

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

## Acknowledgements

We thank T. Bartke (Imperial College London) for providing expression vectors human histones H3 and H4 and associated histone purification protocols; For the SAXS experiments, we thank A. Graziadei for data collection and the ESRF, BAG MX1695 for access to the beamline. We also thank J. Kirkpatrick and the EMBL Proteomics core facility for mass spectrometry quality control of modified histones. We are also grateful to Florent Chuffart for his help in preparing the fluorescence time-course curves corresponding to the FRAP experiments. T.C.R.M was supported by the EMBL Inter-disciplinary Postdoc Programme (EIPOD) under Marie Curie COFUND Actions. The SK laboratory is supported by a grant from 'Foundation pour la Recherche Medicale (FRM)' 'analyse bio-informatique pour la recherche en biologie' program as well as by ANR Episperm3 program and by INCa libre program and by 'Plan Cancer' and 'Fondation ARC'.

## Author contributions

T.C.R.M. and H.G. performed biochemical purifications; T.C.R.M. performed EMSA experiments; B.S. carried out NMR experiments; B.S., T.C.R.M. and T.C. analyzed NMR data; V.R. carried out and analysed ITC experiments; S.K. and S.C. designed and performed FRAP experiments. C.W.M. and T.C. conceived the study. T.C.R.M wrote the manuscript with input from all authors.

## Additional information

**Competing financial interests:** The authors declare no competing financial interests.

