## [Peer Review File · Nature Communications]

Reviewers' comments:

Reviewer #1 (Remarks to the Author):

In this manuscript, Miller et al. used biochemical and biophysical methods to investigate the molecular mechanism of the BET protein Brdt function in chromatin remodeling. Their study is built on an early report by Pivot-Pajot et al. (Mol. Cell 2003), which described acetylation-dependent chromatin reorganization by Brdt through its N-terminal 19 residues interactions with nucleosome. In this study, the authors provided experimental evidence that the N-terminal 19-residue fragment of Brdt interacts with the first bromodomain (BD1) and affects BD1 binding to acetyl-lysine and DNA. Notably, the authors validated that BD1, but not BD2, binds acetyl-lysine and DNA in a bivalent manner, significantly improving Brdt binding to nucleosomes in vitro. The authors also showed that nucleosome structure affects Brdt bromodomain binding to the nucleosome using a chimeric nucleosome-Brdt bromodomain assay. While some new in vitro data of bromodomain/nucleosome interactions are provided in this study, overall, the study falls short in providing a sufficient advance in mechanistic understanding or functional significance of Brdt/nucleosome interactions in chromatin remodeling. Specifically, the authors did not provide any cell biology data, which would be required to demonstrate functional significance of BD1 binding to acetyl-lysine and DNA in a cellular context. The authors also need to answer the question on whether bivalent binding of Brdt to acetylated histone and DNA is really important for the regulation of chromatin modeling, and whether BD1 of Brdt is required for recruiting BD2 binding to nucleosomes. In addition, there are several major issues about specific experiments in the study should be addressed by the authors as follows:

1. The ITC data show that Brdt N-terminus-BD1 and Brdt-BD1 do not bind to unmodified nucleosome, but interact with acetylated nucleosome better than acetylated peptides. The authors need to provide a better discussion for these results. In addition, the EMSA data that show Brdt N-terminus-BD1 and Brdt-BD1 interact with DNA with good affinities, but the authors did not explain why Brdt N-terminus-BD1 and Brdt-BD1 do not bind to the DNA on unmodified nucleosome. The results seem contradict to each other. As such, their conclusion of BET to DNA binding is not validated.
2. Brdt-BD2 binds H3K18acK23ac peptide, but surprisingly, it cannot interact with H3K18acK23ac acetylated nucleosome. Considering that the H3K18acK23ac tail on the nucleosome is flexible and available for the interaction, one possible reason is that the nucleosomes are not properly packaged. The authors need to do control experiments to show that the nucleosomes used are structured and functioning.
3. Supplement Fig. 1c. At line 138, page 7, the authors claimed that Brdt(1) (Brdt-BD1) binding to H3K18acK23ac nucleosome is weak (>100 μ M) based on ITC experiment. The label (N-Brdt (1)) from Fig S1c is not consistent with text description, Brdt (1). Also, the signal noises are too high and heat change upon the titration is very little. The authors need to obtain better ITC data or use other methods to evaluate this interaction.
4. Fig. 2. It is better to overlay Fig. 2d and 2e in order to show nucleosome binding to BrdT does add extra chemical shift of different amino acids in Brdt (1-2) compared to Brdt (1-2) binding to acetylated histones.
5. Fig.3. K_d determined by gel shift is not accurate. The authors should use by ITC or FP assay to measure the K_d . In panel b, why is Brdt binding to long DNA fragment tighter than short DNA fragment, if Brdt interacts DNA nonspecifically and DNA binding region in Brdt is well defined?

6. Fig.4. The authors need to provide NMR spectra of mutant Brdt (1)-DNA interactions in order to prove that mutant Brdt (1) including 2KS and 3KS lose its binding to DNA.
7. Fig.5. Panel b. In addition to gel shift assay, it is better to provide NMR spectra of bromodomain-DNA binding, or use FP to evaluate DNA binding affinity of individual BET bromodomains.
8. Supplement Fig. 6b. The heat dilution in ITC of mutant Brdt(1) I115Y is very big, and this makes the Kd value from ITC not reliable.
9. GST-pull down experiments for Brdt different construct domains with different nucleosomes and chromatins in unmodified and modified fashions.
10. Cell based studies of Brdt with different construct domains to chromatins in unmodified and modified fashions.

Reviewer #2 (Remarks to the Author):

In this study Miller and co-workers describe how the testis-specific BET isoform BRDT binds to DNA and histones via its bromodomain module. The authors employ biophysical methods to measure in solution affinities between BRDT bromodomains with acetylated histone peptides or recombinant nucleosomes. The conclusion of the study is that BRDT-BD1 contains a relatively charged site that can potentiate non-specific interactions with DNA while interacting with acetylated histone peptides. In particular, the authors point out that this type of nucleosome engagement may be different between BET proteins.

Albeit interesting, the study lacks rigorous presentation of data to support the conclusions presented and feels hastily put together. ITC curves look nice; however it is unclear if the interactions are 1:1 or 2:1 - in certain cases (ie figure 1b,c) it is unclear if the titrations are saturated or not. No tables are provided with information on data de-convolution other than the KDs displayed on the figures. The interactions with nucleosomes are intriguing; however there is no information on purity or oligomeric character of these reagents in solution - yet the authors study in solution by analytical ultracentrifugation and small angle X-ray scattering the shape/size of the tandem BRDT bromodomains. BET BRDs (BRD2, BRD3, BRD4, either domain of each) are used in EMSA assays yet there is no experimental section describing what the constructs are and how they were generated handled. A more systematic analysis would have helped convey the message better in my opinion.

It would have been interesting for example to see what the differences between BETs are that lead to BRDT-interactions with DNA. For instance BETs seem to have variable N-termini as well as variable linkers between their BRD modules. How does that affect flexibility and topology of the two domains? The authors state that the N-terminus of BRDT is important for DNA binding and that it affects HSQC signals on the ZA-loop region; however all BETs share the same N-terminal extension beyond the standard BRD module structure; in the case of BRD4(1) several structures are now available and they all share the same PPPP motif found in the terminus of BETs (also true for the first domain of BRDT) which packs next to the ZA-loop. The observation therefore that the N-terminus of BRDT(1) does the same, albeit employing NMR is consistent with previously described structural analyses and does not merit the level of novelty that is attributed to it in the text in my opinion.

Minor comments:

Intro - although the authors introduce BETs as a sub-family, they cite literature that describes BRDT

and tests-specific roles - this should be addressed (either by focusing the text onto BRDT or by linking to BET-family reviews)

The authors mention that chromatin IP and pull downs or in vitro experiments such as ITC have been employed to determine BET/histone interactions. The authors should know that also mass spectrometry has been employed to identify states of acetylated histones recognized by BETs (PMID: 22897906 & 22464331)

If BRDT(1) recognizes non-specifically DNA can the authors demonstrate for example AT- or GC-rich lack of specificity? Or other Widom sequences? Why do the authors choose Widom 601 over other sequences?

The authors mention that they analyzed published structures of human BRDT-BD1 and BRDT-BD2 - are there any references for those? It appears that only BRDT-BD1 has been disclosed (?)

Reviewer #3 (Remarks to the Author):

The Carlomagno and Müller groups present a highly interesting study regarding the interactions between bromodomains (BDs) and acetylated nucleosomes. In particular, they convincingly show that binding experiments using acetylated peptides are not representative for the interaction in a nucleosomal context. In addition, they show that DBs interact with DNA in the nucleosomes. Interaction between BDs and nucleosomes is thus more complex than previously anticipated.

Although the main message of the paper is highly interesting and convincingly presented there are a number of important points that need clarification.

Figure 2: What is the rationale for testing the binding between BD1 and the H3 tail and the BD2 and the H4 tail (and not also the BD1 and the H4 tail and the BD2 and the H3 tail)? In case literature suggest this preference that authors should confirm this experimentally. Especially because the authors find that the interactions between BDs and nucleosomes is different from what is expected.

The authors use two forms of first BD of Brdt, a construct that spans residues 19-143(Brdt) and one that spans residues 1 to 143 (N-Brdt). The authors note that inclusion of the Brdt N-terminal residues increases the affinity for acetylated peptides and nucleosomes. Based on a number of crystal structures of BDs, especially those of Brd4(1) (e.g. 4YH4 or 3JVJ), it is clear that the N-terminal residues of the Brdt BD1 are part of the structure of the domain. Removal of the N-terminal residues from Brdt is thus expected to have an impact on the structure and stability of the domain. In that light is somewhat surprising to me that the authors used the truncated version of the BD in their studies. Regarding that I have the following remarks:

- * The authors should determine the relative stability of N-Brdt(1) and Brdt(1) to ensure that changes in stability of the domain are not the reason for the observed affinity differences.
- * The authors state that "Interestingly, we found that the Brdt N-terminus has a significant impact on Brdt-BD1" (line 129, page 7). Based on available structural data (see above), I would rephrase "Interesting" into "As could be expected".
- * Figure S2: Between N-Brdt and Brdt many NMR resonances are different. The authors only labeled the resonances of residues in the histone binding pocket. However, many more resonances change, as is also expected based on the predicted structure of the N-terminal residues in the domain. The statement "which primarily localize to the ZA-loop that forms a major component of the histone binding pocket" is thus not supported by the current presentation of the data (page 7 line 131) and misleading. A more complete analysis of the chemical shift differences would be required (e.g. on a

per residue basis) to determine if the N terminus indeed has predominantly an influence on the ZA loop.

Figure 2a, S1d S3c: The axis labels are too small and not properly labeled (S1d, S3c).

Figure S3c: and Figure 2c: The overlays are not very clear as resonances of the Brdt(1-2) construct (in black) are not visible under the resonances of the individual domains (in blue or red). For both plots, I suggest showing a spectrum of the free Brdt(1-2) next to the overlay of the three proteins.

Page 9: "Peaks belonging to the linker were predominantly grouped in the middle of the spectrum, suggesting that the linker is unstructured." Is that based on Figure S3c, where many resonances in the center of the proton range appear in the tandem construct, but not in the isolated domains? Or did the authors assign the linker region by any direct means?

Page 9: "Comparable peak intensities for the bromodomains in the linked construct indicate that the two domains rotate independently of each other". In case both BDs would tightly interact with one another one would also expect equal intensities for both domains (as the complete protein would tumble as a single unit). Please clarify.

Page 9: Is Figure S4a the same data as Figure 2C. If so, why are the spectra processed differently in the carbon dimension (different resolution)? The resolution in the carbon dimension in most NMR spectra seems very limited.

Page 9: "BD1 preferentially interacted with H4K5acK8ac, whilst BD2 preferentially interacted with H3K18acK23ac". The addition of one or the other peptide causes CSP in exactly the same resonances in the Brdt(1-2) protein (Figure S4). " Based on that I would conclude that both peptides interact in the same manner with the Brdt(1-2), i.e. that there is no preferential interaction with either domain at all. The authors should clarify this.

Page 9: "The linker and BD2 did not interact with the nucleosomes and remained flexible in solution, showing that tethering of the BD2 to nucleosomes is not sufficient to induce interaction." From what data did the authors conclude that, I couldn't find any data that supports this statement?

Page 10: "Thus, this region appears to interact with DNA on the nucleosome." This is only the case when the truncation does not influence the structure or stability of the domain. Based on the Brd4 structures, the loop around residue 89 will interact with the residues 19 to 27. The reduced affinity could thus very well be an indirect effect.

Page 11: "Therefore, the increased binding affinity between Brdt-BD1 and nucleosomes, when compared to DNA or histone peptides alone, appears to occur largely through the entropic benefits of bivalency" The peptide interacts with an affinity 13 μ M (Figure 1), the DNA with an affinity of 10-52 μ M (Figure 3). Based on that I would expect that the acetylated nucleosomes would bind better than low nM. The actual affinity of 2 μ M shows that the avidity effect is very small. The authors should comment on that.

Page 12: "We performed quantitative EMSA titrations". As the titration is quantitative the authors should extract KD values for the interaction in the presence and absence of the peptide.

Figure 5: Was the N-terminal part (N-terminal of the first helix) included in all proteins structures? If not this part of the domains could change the appearance of the surface potential significantly. Please ensure that all models have the same domain boundaries.

Page 14: "To investigate whether nucleosome structure contributes to target specificity, we produced chimeric nucleosomes in which the acetylated histone H4 tail (H4K5acK8ac) was ligated to the core of histone H3. " Did the authors remove the H3 tail and replace this with the acetylated H4 tail, or were the H3 and H4 tails swapped. The latter is preferable as there the same residues are present in the complex. In the former case the H4 tail is present twice (ones modified, ones unmodified), which can introduce artifacts. The authors should comment on that, especially as this experiment form the basis for the conclusion that "a higher layer of specificity is generated" (page 15).

The authors show that the Brdt-BD2 does not interact with nucleosomes. I wonder if there is any indication about what the biological role of this domain is or if the authors could speculate on the function.

The authors should provide estimates for the errors in the extracted affinities.

Point-by-point Response to reviewers

Firstly, we would like to thank the reviewers for their time and thoughtful comments on our manuscript. We have endeavored to address the issues raised and hope they find the revised version suitable for publication.

Reviewer #1:

In this manuscript, Miller et al. used biochemical and biophysical methods to investigate the molecular mechanism of the BET protein BRDT function in chromatin remodeling. Their study is built on an early report by Pivot-Pajot et al. (Mol. Cell 2003), which described acetylation-dependent chromatin reorganization by BRDT through its N-terminal 19 residues interactions with nucleosome. In this study, the authors provided experimental evidence that the N-terminal 19-residue fragment of BRDT interacts with the first bromodomain (BD1) and affects BD1 binding to acetyl-lysine and DNA.

Response: Surprisingly, the data regarding the N-terminus of BRDT has been raised by each of the reviewers. The N-terminal tail was not intended to be a major feature of the story. We mainly described the role of the N-terminal extension as a means to explain why we use a longer BRDT-BD1 construct, rather than a shorter construct that only comprises the canonical bromodomain fold, which was previously used for crystal structure determination¹. We tried to highlight this in the original manuscript by saying “*Although detailed analysis of this effect is outside the scope of this paper, we note that the BRDT N-terminus is important for the interaction between the bromodomain and nucleosomes and as such we subsequently focus primarily on this longer construct*”.

Because the comparison between the longer and the shorter constructs of BD1 apparently distracted from the main message of the manuscript, in the revised version we no longer refer to the shorter construct at all. We hope that this enhances the readability of the paper.

Notably, the authors validated that BD1, but not BD2, binds acetyl-lysine and DNA in a bivalent manner, significantly improving BRDT binding to nucleosomes in vitro. The authors also showed that nucleosome structure affects BRDT bromodomain binding to the nucleosome using a chimeric nucleosome-BRDT bromodomain assay. While some new in vitro data of bromodomain/nucleosome interactions are provided in this study, overall, the study falls short in providing a sufficient advance in mechanistic understanding or functional significance of BRDT/nucleosome interactions in chromatin remodeling. Specifically, the authors did not provide any cell biology data, which would be required to demonstrate functional significance of BD1 binding to acetyl-lysine and DNA in a cellular context. The authors also need to answer the question on whether bivalent binding of BRDT to acetylated histone and DNA is really important for the regulation of chromatin modeling, and whether BD1 of BRDT is required for recruiting BD2 binding to nucleosomes.

Response: We appreciate the reviewer's honest appraisal of our manuscript and have now performed further experiments to address the issues raised. Most importantly, we have collaborated with the Khochbin group, who are experts in researching the biological roles of BRDT. They performed fluorescence recovery after photobleaching (FRAP) experiments and have demonstrated that bromodomain-mediated DNA binding is important for BRDT's ability to localize to and compact acetylated chromatin in a cellular context. This exciting result highlights the importance of the bivalent binding of BRDT to acetylated chromatin. Furthermore, these results allow us to confirm that the effects we see on modified nucleosomes in our biophysical assays are recapitulated in cells, and are therefore biologically relevant. The results are presented in Figure 6 and Supplementary Figure 9, and are discussed on page 16 of the main text.

In regards to the requirement of BD1 for the recruitment of BD2 to nucleosomes, we'd like to make two points. Firstly, our biophysical data support the conclusion that BD2 cannot interact with acetylated nucleosomes in the manner predicted by peptide studies, whereas BD1 can interact with nucleosomes, and can recruit BD2. Secondly, cell based studies have found that BRDT localization is mainly reliant on BD1, as we note in our manuscript (page 20):

“Previous research has shown that BRDT localization and its ability to compact chromatin in somatic cells are primarily mediated by BRDT-BD1, whilst BRDT-BD2 is, at least in part, dispensable^{1,2}.”

This data has now been supported by own cell biology studies showing that mutation of the acetyl-lysine binding pocket or DNA-binding interface of BD1 alone is sufficient to compromise BRDT's ability to localize to acetylated chromatin. We now comment on this in our results section (page 17):

“the data support our finding that BRDT-BD2 cannot associate with nucleosomes, by demonstrating that BD2 is unable to recruit BRDT to hyperacetylated chromatin in the absence of fully functional BD1.”

The accumulating data, both from ourselves and others, leads us to the conclusion that BRDT-BD1 is required for the recruitment of BD2 to nucleosomes.

1. The ITC data show that BRDT N-terminus-BD1 and BRDT-BD1 do not bind to unmodified nucleosome, but interact with acetylated nucleosome better than acetylated peptides. The authors need to provide a better discussion for these results. In addition, the EMSA data that show BRDT N-terminus-BD1 and BRDT-BD1 interact with DNA with good affinities, but the authors did not explain why BRDT N-terminus-BD1 and BRDT-BD1 do not bind to the DNA on unmodified nucleosome. The results seem contradict to each other. As such, their conclusion of BET to DNA binding is not validated.

Response: The reviewer raises an interesting and important point. To address this apparent discrepancy we have performed EMSA titration experiments with N-BRDT(1) against DNA, unmodified, H3 acetylated, H4 acetylated and double H3 and H4 acetylated nucleosomes (**Fig. 3c**). This experiment confirms that N-BRDT can indeed interact with unmodified nucleosomes in a similar fashion to free DNA, albeit with what appears to be slightly weaker affinity. In addition to Figure 3 we have added the following into the results section (bottom of page 10):

“Our ITC data (Fig. 1b; Supplementary Fig. 2a, b) suggested that N-BRDT(1) required acetylation of H4 to bind to nucleosomes, however binding of N-BRDT(1) to free DNA would suggest that it should also bind non-specifically to nucleosomes, independent of acetylation. To test this, we performed EMSA experiments with 167 bp Widom DNA and unmodified or acetylated nucleosomes. We found that N-BRDT(1) did indeed show weak binding to unmodified nucleosomes in a similar manner seen for DNA. The apparent discrepancy with the ITC result suggests that - due to limitations in the achievable sample concentrations - the interaction could not be detected under our experimental conditions.

As predicted from our ITC experiments (Fig. 1b), N-BRDT(1) binds with a significantly higher affinity and apparent specificity to nucleosomes containing acetylated histone H4 (Fig. 3c). In contrast, BRDT(2) showed no interaction with unmodified or acetylated nucleosomes (Supplementary Fig. 5), supporting our findings from NMR and ITC experiments.

To our knowledge, these data are the first demonstration of DNA binding by a bromodomain and show that BRDT-BD1 interacts non-specifically with DNA and nucleosomes. BRDT therefore has a significantly different mode of interacting with nucleosomes than previously envisaged, and may target bulk chromatin through low affinity, non-specific DNA interactions, priming it to interact tightly and specifically following histone hyperacetylation.”

2. BRDT-BD2 binds H3K18acK23ac peptide, but surprisingly, it cannot interact with H3K18acK23ac acetylated nucleosome. Considering that the H3K18acK23ac tail on the nucleosome is flexible and available for the interaction, one possible reason is that the nucleosomes are not properly packaged. The authors need to do control experiments to show that the nucleosomes used are structured and functioning.

Response: We agree that the addition of our control experiments would remove doubt over the possibility of misfolded nucleosomes causing this effect, and thus would enhance the paper. As such we have included **Supplementary Fig. 1**, which compares octamer refolding and nucleosome reconstitution for each of the modified nucleosomes used in this article. We see no evidence of our modifications interfering with nucleosome assembly. Furthermore, TEV protease is able to efficiently cleave our tagged histone tails (on H3 and H4) in a 2 hour cleavage reaction at 30°C – suggesting both that the tails are accessible and that the nucleosomes are stably folded. We note that BRDT-BD1 is able to specifically interact with these nucleosomes when acetylated on the H4 tail as demonstrated in EMSA, ITC and NMR studies and we therefore have no doubt that they are correctly folded. We have added a line into the text to state: “The histone modifications did not interfere with octamer or nucleosome refolding (**Supplementary Fig. 1**) and therefore were used to investigate how nucleosome structure affects bromodomain binding to histone tails.” (Page 6)

Finally, it should be noted that many excellent labs have been using the same protocols for reconstituting a large array of modified nucleosomes with great success (references include^{3,4}). Therefore it seems that modification by NCL in this way does not impinge significantly on *in vitro* nucleosome reconstitution.

3. Supplement Fig. 1c. At line 138, page 7, the authors claimed that BRDT(1) (BRDT-BD1) binding to H3K18acK23ac nucleosome is weak (>100 μ M) based on ITC experiment. The label (N-BRDT (1)) from Fig S1c is not consistent with text description, BRDT (1). Also, the signal noises are too high and heat change upon the titration is very little. The authors need to obtain better ITC data or use other methods to evaluate this interaction.

Response: It is true that the signal:noise ratio for this experiment is less than we would have hoped, however the low affinity of this interaction and the sample requirements for ITC precluded us obtaining stronger data. We have removed this data from the paper. Evidence from Supplementary Figure 1 shows correct folding of our nucleosomes and accessibility to their tails for TEV cleavage. We believe this data highlights that our nucleosomes are ‘functional’.

4. Fig. 2. It is better to overlay Fig. 2d and 2e in order to show nucleosome binding to BRDT does add extra chemical shift of different amino acids in BRDT (1-2) compared to BRDT (1-2) binding to acetylated histones.

Response: This figure was primarily used to highlight the fact that BD2 does not interact with these acetylated nucleosomes, even when physically linked to them via BRDT-BD1. We have adjusted the figure we previously used, and moved it to the supplement (**Supplementary Fig. 4b**) so that we can highlight how BD2 (within the BRDT(1-2) context) shows binding to the acetylated histones, but shows no interaction with the nucleosomes under identical conditions. In contrast, BD1 (within the BRDT(1-2) context) shows robust binding to both acetylated histones and nucleosomes. The increased complexity of the spectra of BRDT(1-2), in contrast to N-BRDT(1) alone, coupled to the loss of signal of almost all well resolved BD1 peaks when bound to the nucleosomes precludes detailed comparison of the interactions of BD1 with histones vs nucleosomes. Therefore, we feel that the spectra comparing N-BRDT(1) binding to the peptides and nucleosomes better illustrates the points we raise in the article.

5. Fig.3. K_d determined by gel shift is not accurate. The authors should use by ITC or FP assay to measure the K_d . In panel b, why is BRDT binding to long DNA fragment tighter than short

DNA fragment, if BRDT interacts DNA nonspecifically and DNA binding region in BRDT is well defined?

6. Fig.4. The authors need to provide NMR spectra of mutant BRDT (1)-DNA interactions in order to prove that mutant BRDT (1) including 2KS and 3KS lose its binding to DNA.

7. Fig.5. Panel b. In addition to gel shift assay, it is better to provide NMR spectra of bromodomain-DNA binding, or use FP to evaluate DNA binding affinity of individual BET bromodomains.

Response to points 5, 6 & 7: Although we would always like to have more data, we feel that these requests for extra experiments are unwarranted. EMSAs are a well-established method for calculating binding affinities between proteins and nucleic acids (and frequently appear in Nature Communications and other high impact journals). Furthermore, we already thoroughly supplement our EMSA data with NMR and ITC experiments to characterize the interaction interface of interest, as well as how mutation of this interface leads to a decreased affinity for the bromodomains physiological substrate – an acetylated nucleosome. We believe that this data, coupled with our additional cell biology data showing the functional relevance of the DNA binding interface should suffice to support the conclusions made in our article.

The reviewer also asked why the apparent DNA binding affinities detailed in the paper are stronger for longer DNA than shorter DNA. This is to be expected based the concept that a non-specific DNA binding protein will have more potential binding sites on a longer DNA. Many theoretical papers have proposed models to describe this behaviour (examples include ^{5,6}). As BRDT's physiological substrate is an acetylated nucleosome, rather than free DNA, we have not investigated the details of this further.

8. Supplement Fig. 6b. The heat dilution in ITC of mutant BRDT(1) I115Y is very big, and this makes the Kd value from ITC not reliable.

Response: Whilst it is true that the heat of dilution for this experiment appears large, this is a somewhat artificial effect because of the different scale bars presented in the ITC data in Supplementary Fig. 6. The heat of dilution is in fact no larger than the other experiments, however the poor binding affinity means there is little signal from binding. The sole purpose of

this result was to highlight the fact that the I115Y mutant has a significantly weaker binding affinity for acetylated histone tails – a point which is well upheld by this data. Inclusion of the associated error for the calculated binding affinity (**Supplementary Table 1**) highlights the fact that this affinity value is less accurately determined than the other affinities reported in this paper.

9. GST-pull down experiments for BRDT different construct domains with different nucleosomes and chromatin in unmodified and modified fashions.

Response: Although this could be an interesting experiment, we feel that an experiment of this magnitude is out of the scope of this paper. Producing modified nucleosomes is not trivial and very few groups have the capacity to produce libraries of modified nucleosomes for screening. Doing this with chromatin would add another level of complexity. Although the possibility remains that the BRDT bromodomains may interact with a previously unidentified modification with higher specificity, accumulating data, including pull-downs and mass spectrometry, suggest that acetylation is genuine physiological target of the BET bromodomains^{7,8} and that BRDT BD1 specifically has a preference for histone H4 acetylated on K5 and K8^{1,9}.

10. Cell based studies of BRDT with different construct domains to chromatin in unmodified and modified fashions.

Response: We thank the reviewer for this suggestion. To address this point we collaborated with the Khochbin group, which has significant experience in investigating the biological functions of BRDT. We tested WT and mutant BRDT constructs for their ability to associate with and compact hyperacetylated chromatin using well established cell based fluorescent experiments. The results demonstrated that the DNA binding interface of BRDT(1) plays an important role in BRDT's ability to compact TSA-induced hyperacetylated chromatin in cells. Interestingly, mutating the DNA-binding interface of BRDT-BD1 has a similar effect to mutating the acetyl-lysine binding pocket; both of these mutants are compromised in their abilities to localize to hyperacetylated chromatin and then cause compaction. These results are presented in detail in **Figure 6** and discussed in the main text of the paper (page 16) in a new subsection titled: “**BD1-DNA binding is important for chromatin compaction**”.

Reviewer #2 (Remarks to the Author):

In this study Miller and co-workers describe how the testis-specific BET isoform BRDT binds to DNA and histones via its bromodomain module. The authors employ biophysical methods to measure in solution affinities between BRDT bromodomains with acetylated histone peptides or recombinant nucleosomes. The conclusion of the study is that BRDT-BD1 contains a relatively charged site that can potentiate non-specific interactions with DNA while interacting with acetylated histone peptides. In particular, the authors point out that this type of nucleosome engagement may be different between BET proteins.

Albeit interesting, the study lacks rigorous presentation of data to support the conclusions presented and feels hastily put together. ITC curves look nice; however it is unclear if the interactions are 1:1 or 2:1 - in certain cases (ie figure 1b,c) it is unclear if the titrations are saturated or not. No tables are provided with information on data de-convolution other than the KDs displayed on the figures.

Response: We have now included a table to summarise the ITC data more thoroughly (**Supplementary Table 1**), as requested. The interactions we see with nucleosomes reflect a 2:1 stoichiometry, in accordance with the expected interaction of one bromodomain with each of the modified histone tails. The interactions between the bromodomains and peptides all appear to have the 1:1 stoichiometry. Due to the high sample requirements for measuring ITC data on modified nucleosomes we were unable to reach 'complete' saturation in all of our ITC data, however, our data approach saturation and contain sufficient information to obtain a stable fit as necessary for calculating the binding parameters.

The interactions with nucleosomes are intriguing; however there is no information on purity or oligomeric character of these reagents in solution - yet the authors study in solution by analytical ultracentrifugation and small angle X-ray scattering the shape/size of the tandem BRDT bromodomains.

Response: We have now included Supplementary Fig. 1 which demonstrates the purity and oligomeric character of the nucleosomes that we use in our experiments. Further to this, we include SDS-PAGE gels and NMR ^1H spectra obtained to demonstrate the purity and structural integrity of the all bromodomain constructs used in this study (**Supplementary Fig. 11**). All reagents used appear to be monomeric in solution and are highly pure.

BET BRDs (BRD2, BRD3, BRD4, either domain of each) are used in EMSA assays yet there is no experimental section describing what the constructs are and how they were generated handled. A more systematic analysis would have helped convey the message better in my opinion.

Response: We apologise for this oversight in the original version of the article. We have now amended the methods section to describe the constructs used. These proteins were purified using an identical pipeline as the BRDT bromodomain constructs.

It would have been interesting for example to see what the differences between BETs are that lead to BRDT-interactions with DNA. For instance BETs seem to have variable N-termini as well as variable linkers between their BRD modules. How does that affect flexibility and topology of the two domains?

Response: This is an interesting question. It is possible that the variable N-termini (or another region) of the BET proteins contribute to DNA binding *in vivo*, however with our focus on the interactions between the bromodomains and DNA, this is somewhat out of the scope of the paper. In terms of the linker regions, our data support bioinformatic predictions that the linkers are largely disordered and therefore these proteins are predicted to be highly flexible with no fixed orientation between the two bromodomains.

The authors state that the N-terminus of BRDT is important for DNA binding and that it affects HSQC signals on the ZA-loop region; however all BETs share the same N-terminal extension beyond the standard BRD module structure; in the case of BRD4(1) several structures are now available and they all share the same PPPP motif found in the terminus of BETs (also true for the

first domain of BRDT) which packs next to the ZA-loop. The observation therefore that the N-terminus of BRDT(1) does the same, albeit employing NMR is consistent with previously described structural analyses and does not merit the level of novelty that is attributed to it in the text in my opinion.

Response: As stated above, we have decided to remove all data and discussion of the shorter construct from the text. We hope that this enhances the readability of the paper by focusing the results and discussion on the primary findings of the paper; namely, the bromodomain-DNA interaction and its importance for BRDT targeting and chromatin compaction.

Minor comments:

Intro - although the authors introduce BETs as a sub-family, they cite literature that describes BRDT and tests-specific roles - this should be addressed (either by focusing the text onto BRDT or by linking to BET-family reviews)

Response: This has now been addressed in the text (page 3) and referenced as follows:

“The bromodomain and extra-terminal (BET) family (Brd2, 3, 4 and BRDT in human) are multi-functional chromatin effector proteins, whose critical roles in transcription and chromatin biology have made them attractive therapeutic targets for a wide range of malignancies (recently reviewed in ¹⁰⁻¹²).”

The authors mention that chromatin IP and pull downs or in vitro experiments such as ITC have been employed to determine BET/histone interactions. The authors should know that also mass spectrometry has been employed to identify states of acetylated histones recognized by BETs (PMID: 22897906 & 22464331)

Response: We apologise for the omission of these references, they have now been included (page 6).

If BRDT(1) recognizes non-specifically DNA can the authors demonstrate for example AT- or GC-rich lack of specificity? Or other Widom sequences? Why do the authors choose Widom 601 over other sequences?

Response: The Widom 601 sequence was initially chosen as an optimal nucleosome positioning sequence¹³. To compare binding of nucleosomes and DNA we continued to use this sequence in our EMSA experiments to look for DNA binding by the BET bromodomains. The 66 bp and 25 bp sequences are unrelated, but also have a relatively high GC content (64% compared to 59% in Widom DNA). We therefore cannot exclude an AT- or GC-rich specificity from our data. The sequences of DNA samples used in this paper have now also been included in the methods section to clarify this.

The authors mention that they analyzed published structures of human BRDT-BD1 and BRDT-BD2 - are there any references for those? It appears that only BRDT-BD1 has been disclosed (?)

Response: This is an error in the main text. As the figure legend states, the BRDT-BD2 structure shown is a homology model. We have adjusted the main text (page 11) as follows:

“To further characterize the BD1-DNA interaction we analyzed the sequences of human BRDT BD1 and BD2 (Supplementary Fig. 6a), the X-ray crystal structure of BD1 (2RFJ⁷) and a homology model of BD2 (generated using the Phyre2 web server¹⁴)”

Reviewer #3 (Remarks to the Author):

The Carlomagno and Müller groups present a highly interesting study regarding the interactions between bromodomains (BDs) and acetylated nucleosomes. In particular, they convincingly show that binding experiments using acetylated peptides are not representative for the interaction in a nucleosomal context. In addition, they show that DBs interact with DNA in the nucleosomes. Interaction between BDs and nucleosomes is thus more complex than previously anticipated. Although the main message of the paper is highly interesting and convincingly presented there are a number of important points that need clarification.

Figure 2: What is the rationale for testing the binding between BD1 and the H3 tail and the BD2 and the H4 tail (and not also the BD1 and the H4 tail and the BD2 and the H3 tail)? In case literature suggest this preference that authors should confirm this experimentally. Especially because the authors find that the interactions between BDs and nucleosomes is different from what is expected.

Response: Previous work performed in our group characterized the binding preference of the BRDT bromodomains with histone peptides¹ and thus provided our rationale for this follow-up study. Recent data also supports the earlier work that shows BRDT-BD1 is specifically targeted to H4K5acK8ac *in vivo*⁹.

The authors use two forms of first BD of BRDT, a construct that spans residues 19-143(BRDT) and one that spans residues 1 to 143 (N-BRDT). The authors note that inclusion of the BRDT N-terminal residues increases the affinity for acetylated peptides and nucleosomes. Based on a number of crystal structures of BDs, especially those of Brd4(1) (e.g. 4YH4 or 3JVJ), it is clear that the N-terminal residues of the BRDT BD1 are part of the structure of the domain. Removal of the N-terminal residues from BRDT is thus expected to have an impact on the structure and stability of the domain. In that light is somewhat surprising to me that the authors used the truncated version of the BD in their studies. Regarding that I have the following remarks:

* The authors should determine the relative stability of N-BRDT(1) and BRDT(1) to ensure that changes in stability of the domain are not the reason for the observed affinity differences.

Response: In light of the reviewers' comments, experiments and discussion relating to the shorter construct of BD1 have been removed from the paper (please also see comments to reviewer 1 and 2). We feel that the N-terminus has detracted somewhat from the focus of the article and feel that its removal enhances the readability of the paper by focusing attention on the primary story of the paper (the novel identification of a bromodomain-DNA interaction that permits bivalent nucleosome and chromatin binding by BRDT-BD1). This also allows us to add extra data as per the reviewers' requests, without significantly increasing the number of figures or the length of the paper significantly.

Regardless of its removal from the paper, we offer some explanation with regards to our inclusion of this data in the original version of the manuscript:

This construct was initially investigated based on the crystal structure of mouse BRDT(1), but also on the basis that the conserved 'NPPPPE' sequence of the BET bromodomains is not a central feature of the bromodomains in general, but rather a unique component of the sub-family. While the conserved prolines are now present in many of the Brd4(1) structures, to our knowledge the impact of these prolines on the overall bromodomain fold has not been investigated.

To confirm the validity of our findings, we tested the stability of our truncated constructs by thermal denaturation analysis using circular dichroism (CD), as suggested. We attach a figure of these melting curves for the reviewers' reference. Although deletion of 18 or 27 amino acids (and therefore of the NPPPPE sequence) from the constructs reduces the T_m of the constructs to some extent, the proteins are all stably folded at the temperatures of our experiments (EMSA - 4°C; ITC - 20°C). Further, we expressed, purified and tested a construct truncated just prior to the NPPPPE motif ($\Delta 12$). This construct had a very similar thermal denaturation profile to N-BRDT(1), suggesting that the proline residues are important for stabilizing the bromodomain fold. Furthermore, NMR analysis (also included for the reviewers' reference) showed that $\Delta 12$ induced a significant effect on the bromodomain dynamics, increasing the peak intensities for residues throughout the bromodomain. However, despite its commonalities with N-BRDT(1), the spectra of $\Delta 12$ are not identical to those of N-BRDT(1). Loss of the first 12 amino acids of BRDT induced several specific CSPs in the vicinity of the histone binding pocket; we believe that this region is important for regulating the conformation of the acetyl-lysine pocket.

* The authors state that "Interestingly, we found that the BRDT N-terminus has a significant impact on BRDT-BD1" (line 129, page 7). Based on available structural data (see above), I would rephrase "Interesting" into "As could be expected".

Response: Please see the comments above. This section has now been removed.

* Figure S2: Between N-BRDT and BRDT many NMR resonances are different. The authors only labeled the resonances of residues in the histone binding pocket. However, many more resonances change, as is also expected based on the predicted structure of the N-terminal residues in the domain. The statement "which primarily localize to the ZA-loop that forms a major component of the histone binding pocket" is thus not supported by the current presentation of the data (page 7 line 131) and misleading. A more complete analysis of the chemical shift differences would be required (e.g. on a per residue basis) to determine if the N terminus indeed has predominantly an influence on the ZA loop.

Response: Please see the comments above. This section has now been removed to focus on the primary story of the paper. For the reviewer's reference, we note that the $\Delta 12$ construct, which includes the conserved prolines found in the Brd4 BD1 structures, accounts for many of the CSPs we see in response to truncation of the bromodomain, but not all. The residues which are still affected are found in the ZA loop, as illustrated in the overlaid NMR spectra and color-coded structure of BRDT-BD1 included for the reviewer.

Figure 2a, S1d S3c: The axis labels are too small and not properly labeled (S1d, S3c).

Response: All axis labels have now been adjusted as suggested.

Figure S3c: and Figure 2c: The overlays are not very clear as resonances of the BRDT(1-2) construct (in black) are not visible under the resonances of the individual domains (in blue or red). For both plots, I suggest showing a spectrum of the free BRDT(1-2) next to the overlay of the three proteins.

Response: These figures have now been edited as suggested.

Page 9: "Peaks belonging to the linker were predominantly grouped in the middle of the spectrum, suggesting that the linker is unstructured." Is that based on Figure S3c, where many resonances in the center of the proton range appear in the tandem construct, but not in the isolated domains? Or did the authors assign the linker region by any direct means?

Response: The conclusion that the linker is disordered was reached based on the Figure S3c and not on the basis of assignment. We have now edited the text (page 8) to clarify our explanation of the data:

"New resonances, which appeared in the BRDT(1-2) ^1H , ^{15}N HSQC spectrum (Supplementary Fig. 3c), were attributed to the linker; these resonances were predominantly grouped in the middle of the spectrum and have significantly higher intensities, suggesting that the linker is unstructured."

Page 9: "Comparable peak intensities for the bromodomains in the linked construct indicate that the two domains rotate independently of each other". In case both BDs would tightly interact with one-other one would also expect equal intensities for both domains (as the complete protein would tumble as a single unit). Please clarify.

Response: Thank you for highlighting the ambiguity in this statement. We have now clarified this in the text (page 8):

"Comparison of overlaid ^1H , ^{15}N HSQC spectra (Supplementary Fig. 3c) and ^{13}C - ^1H methyl-TROSY spectra (Supplementary Fig. 4a) of BRDT(1-2) with individual N-BRDT(1) and BRDT(2) data showed a good correspondence between peak positions. Our experimental data therefore give no indication of significant changes in the folds of these domains when linked, nor of dimerization between them, as has been proposed for other bromodomains of the BET family¹⁵⁻¹⁷.Comparable peak intensities for the resonances of the bromodomains in the linked construct, when compared to those of N-BRDT(1) and BRDT(2) alone, indicate that the two domains rotate independently of each other."

Page 9: Is Figure S4a the same data as Figure 2C. If so, why are the spectra processed differently in the carbon dimension (different resolution)? The resolution in the carbon dimension in most NMR spectra seems very limited.

Response: Figure 2C has now been moved (S4b) and edited according to comments from reviewer 1 (comment 4.). There are two reasons for the difference in appearance of the spectra.

Firstly, In contrast to BRDT(2) and BRDT(1-2), which were ^{13}C labelled only on the terminal methyl, N-BRDT(1) is also ^{13}C labelled on the carbon neighboring the methyl, causing splitting of the peaks in the ^{13}C dimension when recorded to a high resolution. To simplify the appearance of the spectra we processed them to a resolution that is insufficient to resolve these couplings. As a consequence of the different labelling, the BRDT(2) and BRDT(1-2) spectra appear to have higher resolution in the ^{13}C dimension. Secondly, due to the low concentration and unfavorable tumbling of the molecules when interacting with the nucleosomes, the spectra in these experiments were recorded with an evolution time of 8 ms in the ^{13}C dimension to optimize the signal to noise ratio.

Page 9: "BD1 preferentially interacted with H4K5acK8ac, whilst BD2 preferentially interacted with H3K18acK23ac". The addition of one or the other peptide causes CSP in exactly the same resonances in the BRDT(1-2) protein (Figure S4). " Based on that I would conclude that both peptides interact in the same manner with the BRDT(1-2), i.e. that there is no preferential interaction with either domain at all. The authors should clarify this.

Response: We have now removed this statement for clarity and the text now reads (page 9):

"We then tested how BRDT(1-2) interacts with acetylated peptides and nucleosomes using leucine and valine ^{13}C - ^1H methyl-TROSY NMR. Overlaid spectra of samples containing BRDT(1-2) and acetylated H3 or H4 histone peptides show CSPs for both BD1 and BD2 (Supplementary Fig. 4a, lower panels). The observed CSPs occur in almost identical resonances regardless of the peptide, as would be expected for both peptides targeting the histone binding pockets. These data therefore show that both bromodomains bind to both acetylated histone peptides and highlight the lack of specificity of individual bromodomains for acetylated histone peptides alone."

Page 9: "The linker and BD2 did not interact with the nucleosomes and remained flexible in solution, showing that tethering of the BD2 to nucleosomes is not sufficient to induce interaction." From what data did the authors conclude that, I couldn't find any data that supports this statement?

Response: This statement refers to BRDT(1-2) binding to the double modified nucleosomes (Supplementary Fig.4b). In this experiment, addition of the acetylated nucleosomes caused no

identifiable CSPs in resonances from either the linker or the BD2 domain. Additionally, the linewidths of the BD2 resonances remained unchanged upon addition of the acetylated nucleosomes. The text has now been adjusted to state this more clearly (page 9):

“In contrast to the peptide binding, we found that only BD1 was able to interact with nucleosomes uniformly acetylated on both histones H3 (K18_{ac}K23_{ac}) and H4 (K5_{ac}K8_{ac}) (Supplementary Fig. 4a). Resonances from the linker and BD2 did not show evidence of an interaction with the nucleosomes and therefore appear to have remained flexible in solution. This result shows that tethering of the BD2 to nucleosomes is not sufficient to induce interaction.”

Page 10: "Thus, this region appears to interact with DNA on the nucleosome." This is only the case when the truncation does not influence the structure or stability of the domain. Based on the Brd4 structures, the loop around residue 89 will interact with the residues 19 to 27. The reduced affinity could thus very well be an indirect effect.

Response: Please see the comments above. This section has now been removed.

Page 11: "Therefore, the increased binding affinity between BRDT-BD1 and nucleosomes, when compared to DNA or histone peptides alone, appears to occur largely through the entropic benefits of bivalency" The peptide interacts with an affinity 13 uM (Figure 1), the DNA with an affinity of 10-52 uM (Figure 3). Based on that I would expect that the acetylated nucleosomes would bind better than low nM. The actual affinity of 2 uM shows that the avidity effect is very small. The authors should comment on that.

Response: Although larger increases in binding affinity have been observed when comparing peptide/DNA binding to nucleosome binding (e.g. PWWP of LEDGF/PSIP1 binding to K36me3 nucleosomes^{18,19}), we would not expect this magnitude of change to be generalizable. K36 is positioned close to the exit point of the H3 tail between the DNA encircling the nucleosomes, thus provides a significantly different environment to the H4 K5 and K8 residues we are investigating. We would expect the relative increase in affinity to vary considerably depending on the exact biological context; however we also note several examples in the manuscript (page 19) which suggest that the increase in binding affinity we see is not uncommon:

“Although nucleosome binding can be enhanced by tandem domains bivalently binding nucleosomes (2-3 fold enhancement for bromo-PHD of BPTF²⁰; 3-11 fold enhancement for two PHD fingers of CHD4²¹), this is not the case for BRDT.... a construct encompassing Brd4-BD1 and BD2 was found to bind to nucleosomes acetylated on both histones H3 and H4 with a 2.6-fold increased affinity over Brd4-BD1 alone⁴.”

Page 12: "We performed quantitative EMSA titrations". As the titration is quantitative the authors should extract KD values for the interaction in the presence and absence of the peptide.

Response: This has now been rectified and we include the extracted dissociation constants (page 12).

Figure 5: Was the N-terminal part (N-terminal of the first helix) included in all proteins structures? If not this part of the domains could change the appearance of the surface potential significantly. Please ensure that all models have the same domain boundaries.

Response: The original version of this figure included the domain boundaries as found in the crystal structures (PDB codes in figure legend). Brd4-BD1 contained the N-terminal extension which includes the NPPPPE stretch. In Figure 4, Brd4-BD1 has now been edited and the truncated structure and surface charge of the shorter protein, lacking this extension, is shown in the figure so that all structures are more directly comparable. The other bromodomains retain the boundaries found in the crystal structures, which are similar in all constructs. The figure legend has been adjusted accordingly to highlight the truncation.

Page 14: "To investigate whether nucleosome structure contributes to target specificity, we produced chimeric nucleosomes in which the acetylated histone H4 tail (H4K5acK8ac) was ligated to the core of histone H3. " Did the authors remove the H3 tail and replace this with the acetylated H4 tail, or were the H3 and H4 tails swapped. The latter is preferable as there the same residues are present in the complex. In the former case the H4 tail is present twice (ones modified, ones unmodified), which can introduce artifacts. The authors should comment on that, especially as this experiment form the basis for the conclusion that "a higher layer of specificity is generated" (page 15).

Response: In this experiment the H3 tail was replaced with the acetylated H4 tail, rather than the two tails being swapped. Whilst theoretically we agree that the tail-swap experiment would be ‘cleaner’ we do not see any evidence of our experimental setup leading to artifacts. Critically, our results, and those of others, indicate that BRDT-BD1 does not interact with un-acetylated H4 or H3 tails (in the context of peptides or nucleosomes (**Supplemental Figure 2a,b**)) and therefore this should not interfere. Additionally, including two extra binding sites would only be expected to increase the measured affinity. This would suggest that if there were any involvement of the unmodified H4 tails, we would have underestimated the decrease in the binding affinity caused by moving the acetylated H4 peptide. Thus our conclusion that the position of tail generates extra specificity would hold true.

To ensure that our experimental conditions are clear to the reader, we have added a line to the methods section:

“Chimeric nucleosomes contained chimeric H3-H4K5_{ac}K8_{ac} and wild type H2A, H2B and H4, and thus contained both modified and unmodified H4 tail sequences.”

The authors show that the BRDT-BD2 does not interact with nucleosomes. I wonder if there is any indication about what the biological role of this domain is or if the authors could speculate on the function.

Response: Although we have not looked further into the possible binding partners of BRDT(2), the most likely scenario is that BRDT(2) interacts with an acetylated non-histone protein and recruits it to chromatin as has been previously shown for Brd4(2) recruiting Twist²². We did not wish to speculate too much on specific targets without data, however, we have added the following text to the discussion (page 20) to suggest what we believe to be the most likely binding partners:

“One possible function, which would depend on flexible tethering of BD2 to nucleosomes by BD1, would be the recruitment of an acetylated non-histone protein to chromatin. BRDT would not be unique in this function; other BET proteins are also known to use their bromodomains for interactions with acetylated non-histone proteins²²⁻²⁶.”

Possible candidates for BRDT-BD2 recruitment would be the transition proteins (TPs) and protamines (Prms); both of which are known to be acetylated²⁷⁻²⁹, and both of which are critical for BRDT-mediated post-meiotic genome repackaging during spermatogenesis³⁰. TPs and Prms depend on BRDT for nuclear localization in elongating spermatids, and loss of BD1 leads to their accumulation in the cytoplasm, preventing histone replacement³⁰. Although BRDT has a

nuclear localization sequence, this data indicates that chromatin binding by BRDT-BD1 is required for nuclear retention of BRDT, TPs and Prms and therefore that BRDT interacts, either directly, or indirectly, with TPs and Prms – potentially via BRDT-BD2.”

The authors should provide estimates for the errors in the extracted affinities.

Response: Errors have now been included with all affinities derived from ITC (**Supplementary table 1**) and EMSA experiments (in each of the respective figures).

References

1. Moriniere, J. et al. Cooperative binding of two acetylation marks on a histone tail by a single bromodomain. *Nature* **461**, 664-8 (2009).
2. Pivot-Pajot, C. et al. Acetylation-dependent chromatin reorganization by BRDT, a testis-specific bromodomain-containing protein. *Mol Cell Biol* **23**, 5354-65 (2003).
3. Bartke, T. et al. Nucleosome-interacting proteins regulated by DNA and histone methylation. *Cell* **143**, 470-84 (2010).
4. Nguyen, U.T. et al. Accelerated chromatin biochemistry using DNA-barcoded nucleosome libraries. *Nat Methods* **11**, 834-40 (2014).
5. McGhee, J.D. & von Hippel, P.H. Theoretical aspects of DNA-protein interactions: cooperative and non-co-operative binding of large ligands to a one-dimensional homogeneous lattice. *Journal of molecular biology* **86**, 469-489 (1974).
6. Kelly, R.C., Jensen, D.E. & von Hippel, P.H. DNA "melting" proteins. IV. Fluorescence measurements of binding parameters for bacteriophage T4 gene 32-protein to mono-, oligo-, and polynucleotides. *The Journal of biological chemistry* **251**, 7240-7250 (1976).
7. Filippakopoulos, P. et al. Histone recognition and large-scale structural analysis of the human bromodomain family. *Cell* **149**, 214-31 (2012).
8. LeRoy, G. et al. Proteogenomic characterization and mapping of nucleosomes decoded by Brd and HP1 proteins. *Genome Biology* **13**, 1-18 (2012).
9. Goudarzi, A. et al. Dynamic Competing Histone H4 K5K8 Acetylation and Butyrylation Are Hallmarks of Highly Active Gene Promoters. *Molecular Cell* **62**, 169-180 (2016).
10. Wang, C.-Y. & Filippakopoulos, P. Beating the odds: BETs in disease. *Trends in Biochemical Sciences* **40**, 468-479 (2015).
11. Fu, L.-L. et al. Inhibition of BET bromodomains as a therapeutic strategy for cancer drug discovery. *Oncotarget* **6**, 5501-5516 (2015).
12. Padmanabhan, B., Mathur, S., Manjula, R. & Tripathi, S. Bromodomain and extra-terminal (BET) family proteins: New therapeutic targets in major diseases. *Journal of biosciences* **41**, 295-311 (2016).
13. Lowary, P.T. & Widom, J. New DNA sequence rules for high affinity binding to histone octamer and sequence-directed nucleosome positioning. Edited by T. Richmond. *Journal of Molecular Biology* **276**, 19-42 (1998).
14. Kelley, L.A., Mezulis, S., Yates, C.M., Wass, M.N. & Sternberg, M.J. The Phyre2 web portal for protein modeling, prediction and analysis. *Nat Protoc* **10**, 845-58 (2015).
15. Nakamura, Y. et al. Crystal structure of the human BRD2 bromodomain: insights into dimerization and recognition of acetylated histone H4. *J Biol Chem* **282**, 4193-201 (2007).
16. Garcia-Gutierrez, P., Mundi, M. & Garcia-Dominguez, M. Association of bromodomain BET proteins with chromatin requires dimerization through the conserved motif B. *J Cell Sci* **125**, 3671-80 (2012).
17. Wu, S.Y., Lee, A.Y., Lai, H.T., Zhang, H. & Chiang, C.M. Phospho switch triggers Brd4 chromatin binding and activator recruitment for gene-specific targeting. *Mol Cell* **49**, 843-57 (2013).
18. Eidahl, J.O. et al. Structural basis for high-affinity binding of LEDGF PWWP to mononucleosomes. *Nucleic Acids Res* **41**, 3924-36 (2013).

19. van Nuland, R. et al. Nucleosomal DNA binding drives the recognition of H3K36-methylated nucleosomes by the PSIP1-PWWP domain. *Epigenetics Chromatin* **6**, 12 (2013).
20. Ruthenburg, A.J. et al. Recognition of a mononucleosomal histone modification pattern by BPTF via multivalent interactions. *Cell* **145**, 692-706 (2011).
21. Musselman, C.A. et al. Bivalent recognition of nucleosomes by the tandem PHD fingers of the CHD4 ATPase is required for CHD4-mediated repression. *Proc Natl Acad Sci U S A* **109**, 787-92 (2012).
22. Shi, J. et al. Disrupting the Interaction of BRD4 with Diacetylated Twist Suppresses Tumorigenesis in Basal-like Breast Cancer. *Cancer Cell* **25**(2014).
23. Huang, B., Yang, X.D., Zhou, M.M., Ozato, K. & Chen, L.F. Brd4 coactivates transcriptional activation of NF-kappaB via specific binding to acetylated RelA. *Mol Cell Biol* **29**, 1375-87 (2009).
24. Gamsjaeger, R. et al. Structural basis and specificity of acetylated transcription factor GATA1 recognition by BET family bromodomain protein Brd3. *Mol Cell Biol* **31**, 2632-40 (2011).
25. Lamonica, J.M. et al. Bromodomain protein Brd3 associates with acetylated GATA1 to promote its chromatin occupancy at erythroid target genes. *Proc Natl Acad Sci U S A* **108**, E159-E168 (2011).
26. Schröder, S. et al. Two-pronged Binding with Bromodomain-containing Protein 4 Liberates Positive Transcription Elongation Factor b from Inactive Ribonucleoprotein Complexes. *Journal of Biological Chemistry* **287**, 1090-1099 (2012).
27. Gupta, N., Madapura, M.P., Bhat, U.A. & Rao, M.R. Mapping of Post-translational Modifications of Transition Proteins, TP1 and TP2, and Identification of Protein Arginine Methyltransferase 4 and Lysine Methyltransferase 7 as Methyltransferase for TP2. *J Biol Chem* **290**, 12101-22 (2015).
28. Pradeepa, M.M. et al. Acetylation of transition protein 2 (TP2) by KAT3B (p300) alters its DNA condensation property and interaction with putative histone chaperone NPM3. *J Biol Chem* **284**, 29956-67 (2009).
29. Brunner, A.M., Nanni, P. & Mansuy, I.M. Epigenetic marking of sperm by post-translational modification of histones and protamines. *Epigenetics Chromatin* **7**, 2 (2014).
30. Gaucher, J. et al. Bromodomain-dependent stage-specific male genome programming by Brdt. *EMBO J* **31**, 3809-20 (2012).

For Reviewers' Reference

(a) Overlaid thermal denaturation circular dichroism (CD) spectra (222 nm) of BRDT-BD1 truncation constructs (data shown is the mean normalised ellipticity from two experiments). A decrease in thermal stability of the BRDT(1) and N-BRDT(1)-Δ27 construct can be seen above 40°C, however there is no significant difference in the thermal stability of these proteins at 20°C. A coomassie stained SDS-PAGE gel of the purified BRDT constructs is shown to the right. (b) Overlaid ¹H, ¹⁵N HSQC spectra of N-BRDT(1) and N-BRDT(1)-Δ12 shown beside a homology model of human BRDT-BD1 (generated using the Phyre2 web server) that has been coloured to highlight residues showing CSPs between the two different constructs. CSPs cluster around the ZA-loop.

REVIEWERS' COMMENTS:

Reviewer #1 (Remarks to the Author):

With the new data from in vitro experiment as well as functional studies in cells, the authors have addressed most of my concerns, strengthening their conclusion. In addition, the readability of manuscript was also significantly improved through reorganizing data and figures. Therefore, I recommend this manuscript for publication.

Reviewer #3 (Remarks to the Author):

The revised version of the manuscript significantly improved the presentation and quality of the data. I would recommend acceptance of the manuscript when some minor points are considered.

First: Not all binding measurements include errors (especially the ITC data). These errors are important to be able to judge the significance of the differences between various constructs.

Second: There is an error in the caption of Figure S4B: there is no red spectrum in the panels.

Point-by-point Response to reviewers

Firstly, we would like to thank the reviewers again for their time spent reviewing our manuscript. We were glad to hear they now support publication of our article and have addressed the two minor issues raised, as detailed below.

Reviewer #1:

With the new data from in vitro experiment as well as functional studies in cells, the authors have addressed most of my concerns, strengthening their conclusion. In addition, the readability of manuscript was also significantly improved through reorganizing data and figures. Therefore, I recommend this manuscript for publication.

Response: Thank you for your feedback.

Reviewer #3:

The revised version of the manuscript significantly improved the presentation and quality of the data. I would recommend acceptance of the manuscript when some minor points are considered.

- First: Not all binding measurements include errors (especially the ITC data). These errors are important to be able to judge the significance of the differences between various constructs.

Response: For our EMSA binding studies, these errors are shown alongside the calculated binding affinities within the figures (Fig. 3 and Supplementary Fig. 7).

The errors for the ITC binding affinities are shown in Supplementary Table 1, in the column with the calculated binding affinities.

- Second: There is an error in the caption of Figure S4B: there is no red spectrum in the panels.

Response: We have edited the figure legend error in Supplementary Figure 4B.